# Scaling Laws with Vocabulary: Larger Models Deserve Larger Vocabularies

**Chaofan Tao**[1,2]   **Qian Liu**[2†]   **Longxu Dou**[2†]   **Niklas Muennighoff**[3,4]
**Zhongwei Wan**[5]   **Ping Luo**[1]   **Min Lin**[2]   **Ngai Wong**[1†]

[1]The University of Hong Kong   [2]Sea AI Lab   [3]Contextual AI
[4]Stanford University   [5]The Ohio State University

## Abstract

Research on scaling large language models (LLMs) has primarily focused on model parameters and training data size, overlooking the role of vocabulary size. We investigate how vocabulary size impacts LLM scaling laws by training models ranging from 33M to 3B parameters on up to 500B characters with various vocabulary configurations. We propose three complementary approaches for predicting the compute-optimal vocabulary size: IsoFLOPs analysis, derivative estimation, and parametric fit of the loss function. Our approaches converge on the conclusion that the optimal vocabulary size depends on the compute budget, with larger models requiring larger vocabularies. Most LLMs, however, use insufficient vocabulary sizes. For example, we predict that the optimal vocabulary size of Llama2-70B should have been at least 216K, 7 times larger than its vocabulary of 32K. We validate our predictions empirically by training models with 3B parameters across different FLOPs budgets. Adopting our predicted optimal vocabulary size consistently improves downstream performance over commonly used vocabulary sizes. By increasing the vocabulary size from the conventional 32K to 43K, we improve performance on ARC-Challenge from 29.1 to 32.0 with the same 2.3e21 FLOPs. Our work highlights the importance of jointly considering tokenization and model scaling for efficient pre-training. The code and demo are available at `https://github.com/sail-sg/scaling-with-vocab` and `https://hf.co/spaces/sail/scaling-with-vocab-demo`.

## 1 Introduction

Large language models (LLMs) achieve remarkable performance by pre-training on vast text corpora using massive computational resources [47]. Extensive prior work on LLMs has focused on deriving so-called scaling laws: a set of empirical formulas to predict how model performance scales, mainly as computing floating-point operations (FLOPs), model parameters, and quantity of training data change [30, 26, 66, 2, 44, 58]. These works show that power-law fits can effectively predict language modeling loss and by extension downstream performance [23, 55]. However, these scaling laws usually disregard the impact of the vocabulary size. For example, in Kaplan et al. [30] only non-vocabulary parameters are considered in their predictive formula. This negligence has resulted in substantial variability in the vocabulary size of current LLMs. For instance, Llama2-7B employs a vocabulary size of 32K [70], while Gemma-7B [67] adopts a much larger vocabulary size of 256K despite both having a similar number of total parameters. This variability in vocabulary sizes across LLMs raises the research question: *What is the compute-optimal vocabulary size for a LLM?*

---

[†]Corresponding authors. The project was done during Chaofan Tao's internship at Sea AI Lab. For more information, please contact `cftao@connect.hku.hk` and `liuqian.sea@gmail.com`.

38th Conference on Neural Information Processing Systems (NeurIPS 2024).

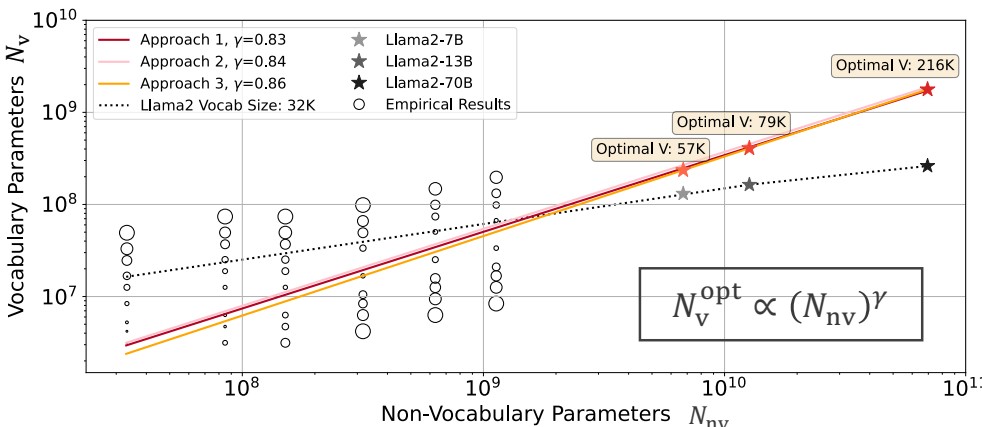

Figure 1: The relationship between non-vocabulary parameters $N_{\mathrm{nv}}$ and the corresponding optimal vocabulary parameters $N_{\mathrm{v}}^{\mathrm{opt}}$ follows a power law, where $N_{\mathrm{v}}^{\mathrm{opt}}$ should be scaled slower than $N_{\mathrm{nv}}$ as $\gamma < 1$. Empirical results align with predictions of our proposed approaches, with larger circles indicating higher loss values. Here $V$ refers to the vocabulary size i.e. the number of distinct tokens.

The vocabulary size affects performance non-trivially. Intuitively, the optimal vocabulary size should neither be too large nor small. A larger vocabulary size improves tokenization fertility, i.e., splitting sentences into fewer tokens, thereby improving the tokenization efficiency. Additionally, a larger vocabulary enables the model to capture a wider range of concept. However, the risk of under-fitting for rare tokens increases with larger vocabulary sizes, especially in the data-constrained regime [44, 72]. Thus, the optimal vocabulary size needs to be determined by taking the training data and the non-vocabulary parameters into account.

In this paper, we show that the effect of vocabulary on scaling laws has been underestimated, and we quantify the effect to derive a prediction for the optimal vocabulary size. We first introduce a normalized loss formulation to ensure a fair comparison across models with varying vocabulary sizes. Utilizing the normalized loss function, we analyze and discuss the underlying rationale behind the existence of an optimal vocabulary size, which depends on the available computational budget.

To predict the optimal vocabulary size given a compute budget, we propose three approaches. **Approach 1 (Estimating power laws via IsoFLOPs)**: We pre-train models with non-vocabulary parameters ranging from 33M to 1.13B, with groups of models that share the same FLOPs ("IsoFLOPs") but varying vocabulary configurations. Then, we fit power laws relating FLOPs to non-vocabulary parameters, vocabulary parameters, and training data, respectively. Our analysis reveals that the optimal vocabulary parameters exhibit a power-law growth with respect to the computational budget, however, at a slower rate than non-vocabulary parameters, as shown in Figure 1. **Approach 2 (Derivative-based Estimation)**: We introduce a derivative-based method that estimates the optimal vocabulary size by using the derivative of FLOPs w.r.t. the vocabulary size and finding the corresponding zero solution. **Approach 3 (Parametric Fit of Loss Formula)**: We modify Chinchilla scaling laws [26] to incorporate vocabulary and fit the resulting formula on our models to predict the normalized loss function based on non-vocabulary parameters, vocabulary parameters, and the amount of training characters jointly. While the prior two approaches are limited to compute-optimal settings, this approach also allows us to determine the optimal vocabulary when the allocation is suboptimal i.e. the model parameters are either trained for too many tokens ("overtrained") or for too few tokens ("undertrained"). Overtraining is very common [23], such as Llama 2 7B [70] which was trained for 2 trillion tokens, significantly more than the compute-optimal allocation of a 7 billion parameter model of around 150B tokens.

As shown in Figure 1, we observe that the relationship between non-vocabulary parameters $N_{\mathrm{nv}}$ and their correspondng optimal vocabulary parameters $N_{\mathrm{v}}^{\mathrm{opt}}$ follows a power law, according to all of our approaches. Our prediction also suggests that vocabulary parameters should be scaled slower than non-vocabulary parameters, i.e., $N_{\mathrm{v}}^{\mathrm{opt}} \propto N_{nv}^{\gamma}$ where $\gamma \approx 0.83 < 1$. Nevertheless, most of existing LLMs [33, 81, 67, 41, 4, 25, 12, 7, 46, 84] neglect the importance of vocabulary and allocate less vocabulary parameters than the suggestions, shown in Figure 2. Note that we assume that

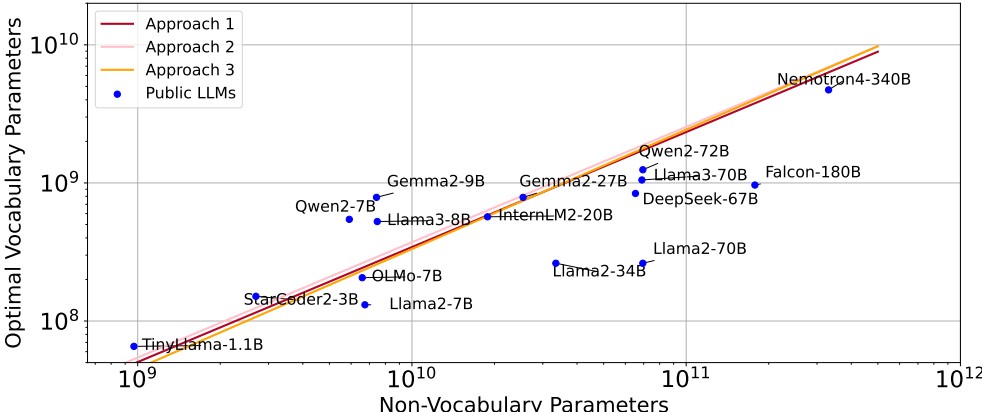

Figure 2: Vocabulary parameters of popular LLMs and predicted optimal vocabulary parameters at a **compute-optimal number of training tokens**. Most current LLMs have suboptimal vocabulary parameters due to vocabulary sizes, which are smaller than the predicted optimal values. Among the current models, StarCoder2-3B, OLMo-7B, InternLM2-20B, and Gemma2-27B have vocabulary sizes that come closest to the optimal allocation for their respective model sizes.

the amount of training data for these models is optimally distributed according to Hoffmann et al. [26]. Considering that several LLMs are trained on substantially more data than optimal ones (e.g., Llama2), the optimal vocabulary sizes would likely be larger than currently estimated.

Finally, we empirically verify our predictions on models with 3B parameter models. By using our approach to predict the expected vocabulary size in various practical cases when (1) the training data is insufficient ("undertraining"); (2) the training data is equally scaled with the model parameters, following the Chinchilla laws ("compute-optimal training") [26]; (3) the training data is overly sufficient like in Llama [70] ("overtraining"). The results show that models with our suggested vocabulary sizes steadily outperform baselines adopting commonly used vocabulary configurations under the same FLOPs budget. Our research underscores the overlooked importance of vocabulary and the need to jointly consider the vocabulary size, model parameters, and training data for effective scaling.

## 2 Preliminary

In this section, we first present a general formulation of a commonly used scaling law, and then demonstrate how to modify it to incorporate the vocabulary.

### 2.1 Scaling law

Scaling laws consider a computational budget, $C$, which is measured in FLOPs. The goal is to optimally allocate the compute budget to model parameters $N$ and the number of training tokens $D$ [30, 6, 26, 44]. It can be formulated as:

$$(N^{\text{opt}}, D^{\text{opt}}) = \arg\min_{N, D} \mathcal{L}(N, D) \quad \text{s.t. FLOPs}(N, D) = C, \quad (1)$$

Following Radford et al. [51], the loss function is typically the language modeling loss when evaluating language models, which can be written as:

$$\mathcal{L} = -\frac{1}{T} \sum_{i=1}^{T} \log p(w_i | w_{1:i-1}, V), \quad (2)$$

where $p(w_i | w_{1:i-1}, V)$ is the output probability of word $w_i$ given the context $w_{1:i-1}$ and the tokenizer with vocabulary size $V$. Generally, the lower $\mathcal{L}$ indicates better performance of the language model. However, due to its dependency on $V$, $\mathcal{L}$ cannot be used to compare language models with different vocabulary sizes. Thus, we propose an adaptation later in §2.2. Fitting scaling laws generally requires various models trained for different configurations [23]. A common approach is to select several

compute budgets and train models with varying $N$ and $D$ for each budget to find the best one, i.e. the one with the lowest loss ("IsoFLOPs") [26]. Using fitting techniques we can then estimate a function that maps from the compute budget to the optimal allocation to $N$ and $D$.

## 2.2 Scaling law with vocabulary

As prior work generally assumes the vocabulary size to be fixed, we cannot adopt the attributes in their scaling laws and their evaluation metric directly. Thus, we detail several considerations that allow us to investigate vocabulary scaling laws.

**Attributes**    Scaling laws commonly deal with the attributes, model parameters ($N$) and number of training tokens ($D$) [26, 44]. We adapt them for our analysis in the context of vocabulary size. (1) We break down the total model parameters ($N$) into non-vocabulary ($N_{\mathrm{nv}}$) and vocabulary parameters ($N_{\mathrm{v}}$). To understand the importance of vocabulary parameters, we isolate them from other model parameters, where $N = N_{\mathrm{nv}} + N_{\mathrm{v}}$. We use $N_{\mathrm{v}} = Vd$ to represent both the vocabulary parameters in the output layer [1]. Notably, to change $N_{\mathrm{v}}$ we only vary the vocabulary size $V$ and take the embedding dimension $d$ as given based on $N_{\mathrm{nv}}$ empirically, see §A.7.2 for details. This is based on the observation by Kaplan et al. [30] that the performance of models with varying depth-to-width ratios converges to a single trend. We also provide further analysis about why we break down the model parameters from the perspective of parameter growing in §A.6. (2) We measure data not in tokens ($D$) but in training characters ($H$). The number of tokens depends on the vocabulary of the tokenizer. By studying training characters, we can better see how the data volume affects the performance regardless of different vocabulary sizes.

**Mapping from training characters ($H$) to tokens ($D$)**    As detailed above we measure training data in training characters ($H$). Nonetheless, to connect our findings with existing studies on scaling laws [26, 44], we need to be able to map from $H$ to $D$. This mapping is the tokenizer's compression ratio which can be computed via $D/H$. The more tokens the tokenizer needs to represent $H$, the larger $D$, and thus it compresses less. We develop a simple function $f(V)$ to estimate this ratio solely from the chosen vocabulary size, $V$. Specifically, we find that a quadratic function on the logarithmic value of $V$ achieves accurate predictions:

$$f(V) = a \log^2(V) + b \log(V) + c \tag{3}$$

By fitting several tokenizers with $V$ ranging from $1K$ to $1024K$, we obtain $a = 0.0064$, $b = -0.1581$ and $c = 1.2047$. We find that our function accurately predicts the compression ratio with a low relative mean square error (RMSE) and a high coefficient of determination ($R^2$). In §A.9, we visualize fitting results and show that our approximation works with different tokenizers and is robust to different $V$. For all our main experiments, we use the BPE algorithm for tokenization [59].

**Vocabulary-insensitive loss**    To fairly assess models that vary in $V$, the commonly used language model loss in Equation 2 is inappropriate. Models trained with larger $V$ naturally have a higher loss, as there are more possibilities in the vocabulary to predict. However, this does not mean that the model is worse. Thus, we need to normalize the loss with respect to the vocabulary size. We reformulate the unigram-normalized metric [54] as a loss function. Suppose we have a $T$-length sequence $w_{1:T}$, we design the unigram-normalized language model loss as:

$$\mathcal{L}_u = -\frac{1}{T} \sum_{i=1}^{T} \log \frac{p(w_i|w_{1:i-1}, V)}{p(w_i|V)}, \tag{4}$$

where $p(w_i|V)$ is the frequency of word $w_i$ in the tokenized corpus, given the tokenizer with vocabulary size $V$. The loss indicates the improvement in probability that a context-aware language model offers over a unigram model without context, allowing us to assess the language model's efficacy. Based on theory from prior work [54], the normalized loss $\mathcal{L}_u$ remains consistent for a given model with a fixed non-vocabulary component across different vocabulary sizes. The difference of $\mathcal{L}_u$ comes from the ability of the language model itself. Compared with $\mathcal{L}$, the value of $\mathcal{L}_u$ is much smaller and can be negative as $\mathcal{L}_u$ adds a negative term $\frac{1}{T} \sum_{i=1}^{T} \log p(w_i|V)$. One may also

---

[1]Vocabulary parameters typically encompass both the word embedding layer and the output layer. In this paper, for clarity and analytical simplicity, we employ $Vd$ rather than $2Vd$ to represent the vocabulary parameters. This choice is predicated on empirical observations: the main computational burden, as measured in FLOPs, is associated with the output layer, but not the word embedding layer.

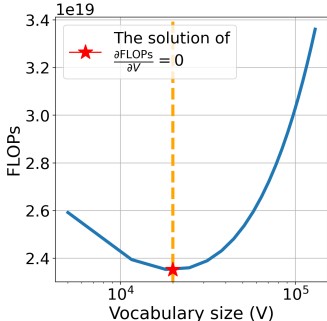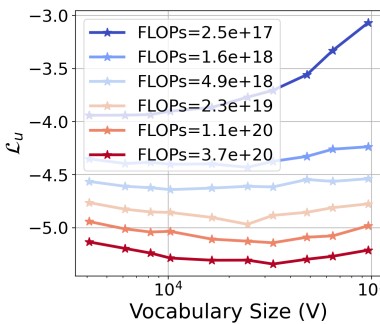

Figure 3: **Left:** FLOPs curve with various vocabulary sizes, assuming all configurations achieve a fixed loss. There exists an optimal vocabulary size that minimizes FLOPs. **Right:** Loss curves with various vocabulary sizes given different FLOP budgets. For each budget there exists an optimal vocabulary size that minimizes loss. As the FLOP budget increases this optimal vocabulary size increases (shifts to the right).

employ the average bits per character (BPC), a common metric for text compression [27], as the vocabulary-insensitive loss. The only difference lies in the normalization. BPC represents the raw per-character language model loss over the corpus, while our $\mathcal{L}_u$ is equivalent to the per-character language model loss normalized by the frequency of each character. In practice, we find that the metric BPC and $\mathcal{L}_u$ show a significant positive correlation, which experimentally validated our statement, as detailed in the §A.5.

## 3 Analysis: Why the optimal vocabulary size is bounded by compute

**Analysis 1: The perspective of fixed normalized loss**     According to Kaplan et al. [30], the FLOPs ($C$) of a Transformer model can be estimated as $C \approx 6ND$, which can be re-written as:

$$C \approx 6ND \approx 6(N_{\mathrm{nv}} + Vd)Hf(V), \tag{5}$$

where $N = N_{\mathrm{nv}} + N_{\mathrm{v}}$ and $D = Hf(V)$ based on §2.2. The reasons why model performance first increases and then decreases as the vocabulary size grows are: (1) At small $V$, increasing the vocabulary size easily improves tokenization fertility from $f(V)$. Subsequently, more characters can be learned from the model with a fixed number of tokens, thereby improving model performance. (2) At very large $V$, the gain from tokenization fertility decreases, while the parameters from expanding the vocabulary cannot be adequately trained with limited data, which leads to a decline in model performance. We present an expanded derivation in §A.1, and show how the corresponding FLOPs change with the vocabulary size in Figure 3 (left).

**Analysis 2: The perspective of fixed FLOP budget**     Given a fixed FLOPs budget, we isolate the FLOPs and investigate how the vocabulary influences the loss. For ease, we train models with fixed $N_{nv}$ and different vocabulary sizes for the same steps, and then we use interpolation to predict the loss when FLOPs reaches the budget given the observed FLOPs and loss points. For each budget, we adopt a group of models with similar total parameters and vary vocabulary sizes. In Figure 3 (right) we plot the relationship between the loss w.r.t. the vocabulary size. It reveals that the vocabulary corresponding to the lowest point on the loss curve increases as the FLOPs budget increases. This suggests that with more computational resources, LLMs can harness larger vocabularies to reduce loss. However, merely expanding the vocabulary does not always lower the loss. For a fixed FLOPs budget, the loss initially decreases with the increase in vocabulary and then starts to rise, indicating that an optimal point exists for the vocabulary.

## 4 Estimating the optimal vocabulary size

In this section, we describe three complementary approaches to estimate the optimal vocabulary size.

## 4.1 Approach 1: Estimating power laws via IsoFLOPs

We define 6 groups of models with $N_{\mathrm{nv}}$ ranging from 33M to 1.13B. Within each group, we solely vary the vocabulary size $V$ from $4K$ to $96K$, and evaluate different models under the same FLOPs budget. We evaluate the normalized loss $\mathcal{L}_u$ on a held-out validation dataset. This approach allows us to directly answer the question: For a given FLOPs budget, what is the optimal allocation to non-vocabulary parameters, vocabulary parameters, and training data?

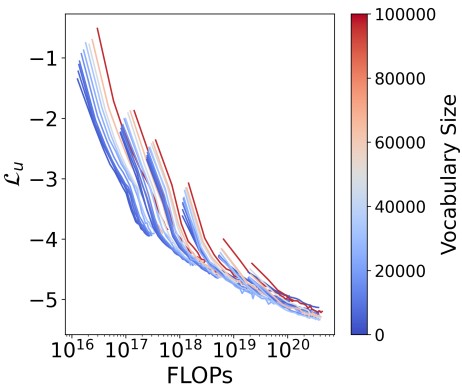

Figure 4: Training curves of the experiments used in Approach 1 (§4.1) and Approach 3 (§4.3). We train models with the non-vocabulary parameters fixed and vocabulary sizes varying from 4K to 96K.

**Setup** Given a certain $N_{\mathrm{v}}$, the embedding dimension $d$ is fixed, thus $N_{\mathrm{v}}$ increases as $V$ increases. For all experiments, we uniformly sample the training data from different domains in the SlimPajama dataset [61]. All other hyperparameters are fixed with more details in §A.7.

**Fitting** We select data points with the minimum $\mathcal{L}_u$ for each FLOP budget, with all runs visualized in Figure 4. These points are the compute-optimal allocation to $(N_{\mathrm{nv}}, N_{\mathrm{v}}, H)$. Following Kaplan et al. [30] and Hoffmann et al. [26], we hypothesize that the optimal vocabulary parameters $N_{\mathrm{v}}$ meet a power law w.r.t. the FLOPs $C$, just like the non-vocabulary parameters and the amount of training data. Specifically, $N_{\mathrm{nv}} = k_1 C^{\alpha_1}, N_{\mathrm{v}} = k_2 C^{\alpha_2}$ and $H = k_3 C^{\alpha_3}$. As model size and training data should be scaled equally for compute-optimal training [26], we set $\alpha_1 = \alpha_3$. As our new attribute $V$ signifi-

cantly increases the number of possible experimental configurations, we employ interpolation across data points to obtain more configurations cheaply. The details of the fitting are in §A.7.4.

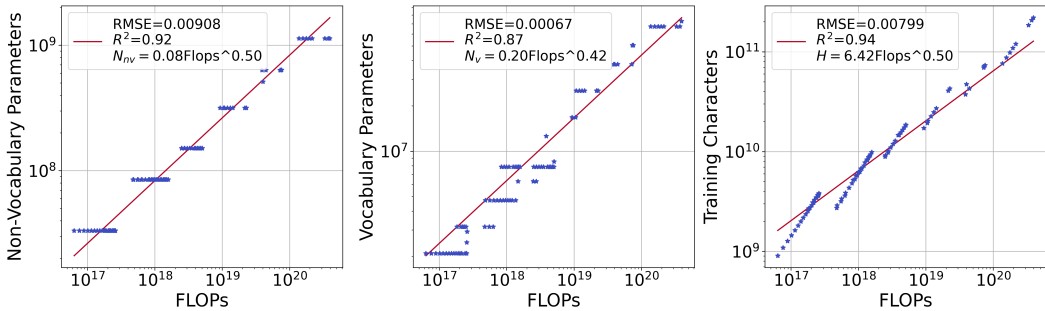

Figure 5: Fitting results of the Approach 1. Blue stars denote the selected data points where the combination $(N_{\mathrm{nv}}, N_{\mathrm{v}}, H)$ reaches the lowest loss given various FLOPs budgets. We find power law fits with respect to the optimal non-vocabulary parameters, vocabulary parameters, and the number of training characters, respectively.

**Results and Usage** In Figure 5, we display the fitted power laws: $N_{\mathrm{nv}} = 0.08 * C^{0.50}$, $N_{\mathrm{v}} = 0.20 * C^{0.42}$ and $H = 6.42 * C^{0.50}$, where $C$ is the FLOPs budget. The low RMSE and high $R^2$ values indicate the strength of our fit. Given a certain FLOPs budget, we can utilize the aforementioned relationships to obtain the optimal allocation $(N_{\mathrm{nv}}, N_{\mathrm{v}}, H)$. We also draw the following conclusions: **(1) LLMs are data-hungry.** Compared to the non-vocabulary parameters $N_{\mathrm{nv}}$, practitioners should allocate more compute to the training data [80, 44]. **(2) Vocabulary parameters scale in a power-law relation with FLOPs** ($N_{\mathrm{v}} \propto C^{0.42}$)**.** As models become more computationally intensive, a larger vocabulary enhances the model's ability to understand a more diverse array of text, and thus the vocabulary size is critical to scaling. **(3) Vocabulary parameters $N_{\mathrm{v}}$ should be scaled slower than non-vocabulary parameters $N_{\mathrm{nv}}$.** This difference can be seen in their power law exponents, i.e. $\gamma = 0.42/0.50 = 0.84 < 1$. We hypothesize the reason is that: once a sufficiently rich embedding space is present via a large vocabulary, it is more critical to scale non-vocabulary parameters to learn the intricate syntactic and semantic structures of language via Transformer blocks.

## 4.2 Approach 2: Derivative-based fast estimation

We propose an alternative approach leveraging insights from the estimation of the FLOPs itself. Prior work [26, 30] usually considers a fixed compute budget in FLOPs and then aims to minimize loss by finding the optimal allocation to model parameters $N$ and training tokens $D$. Here we flip this recipe on its head following recent work [57]. We aim to find the minimum FLOPs to achieve a certain loss $\mathcal{L}_u(N_{\text{nv}}, V, H) = \ell$ through optimal allocation of the vocabulary size $V$:

$$V = \underset{V|\mathcal{L}_u(N_{\text{nv}}, V, H) = \ell}{\arg\min} C(N_{\text{nv}}, N_v, H). \tag{6}$$

By computing the minimum point of FLOPs $C$ with respect to $V$ via derivative:

$$\frac{\partial C}{\partial V} = 6H\left[(N_{\text{nv}} + Vd)\frac{2a\log(V) + b}{V} + \left[a(\log(V))^2 + b\log(V) + c\right]d\right], \tag{7}$$

we can estimate the optimal $V$ under the assumption that it can achieve a certain loss $\mathcal{L}_u(N_{\text{nv}}, V, H) = \ell$. The parameters $a$, $b$ and $c$ can be easily obtained from building $f(V)$ (§2.2). In theory, as long as the non-vocabulary parameters $N_{\text{nv}}$ are provided, $V$ can be numerically searched via the solution of $\frac{\partial C}{\partial V} = 0$. More details are in §A.1.

**Usage** When the compute allocation is near optimal, the loss exhibits a power-law relationship with respect to the FLOPs budget, as described by the scaling law [30]. This relationship allows us to use FLOPs as a reliable proxy for observing the scaling behavior of the optimal vocabulary parameters. In practice, we can first determine an empirically optimal vocabulary size in a low-cost setting (e.g., finding the compute-optimal vocabulary parameters on a small model). Then, we can scale the optimal vocabulary parameters proportionally based on $\gamma$. Specifically, we obtain a set of derivative-optimal vocabulary parameters $N_v$ for different non-vocabulary parameters $N_{nv}$, represented as $\{(N_{\text{nv}}^i, N_v^i)|i = 1, \cdots, n\}$. We then fit the relationship between $N_{\text{nv}}$ and $N_v$ using the power-law function $N_v \propto N_{\text{nv}}^\gamma$. This results in the scaling equation: $N_v/N_v^0 = (N_{\text{nv}}/N_{\text{nv}}^0)^\gamma$ where $N_{\text{nv}}^0$ is a small model (e.g., 33M), and $N_v^0$ is the searched optimal vocabulary parameter. By combining the $\gamma$ from the derivative and the empirical solution on a small model, we can estimate the optimal vocabulary by:

$$N_v^{\text{opt}} = N_v^0 * (\frac{N_{\text{nv}}}{N_{\text{nv}}^0})^\gamma,$$

where the scaling proportion $\gamma = 0.83$ after our fitting. Consistent with the observation in Approach 1, we find that non-vocabulary parameters should be scaled **faster** than vocabulary parameters to achieve an optimal allocation.

## 4.3 Approach 3: Parametric fit of loss formula

Finally, we directly predict the loss given the non-vocabulary parameter, vocabulary parameter and the amount of training characters. Then, the optimal vocabulary configuration can be predicted by finding the minimum point of loss with respect to the vocabulary. Following a classical risk decomposition used in Hoffmann et al. [26], we design the vocabulary-dependent loss formula as:

$$\mathcal{L}_u = -E + \frac{A_1}{N_{\text{nv}}^{\alpha_1}} + \frac{A_2}{N_v^{\alpha_2}} + \frac{B}{D^\beta}, \tag{8}$$

where $D = Hf(V)$. The first term captures the normalized loss for an ideal generative process. The subsequent terms reflect the effect of the non-vocabulary parameters, vocabulary parameters, and the number of training data on the loss, respectively. $E, A_1, A_2, B, \alpha_1, \alpha_2, \beta$ are learned parameters.

**Fitting** We use the points $(N_{\text{nv}}, N_v, H)$ collected for experiments in §4.1. Note that we do not only consider the points with the lowest loss for each FLOP budget as we want to predict loss for any combination of $(N_{\text{nv}}, N_v, H)$. We add the constraint $\alpha_1 = \beta$ following Muennighoff et al. [44]. We also filter out points with very small FLOPs following Hoffmann et al. [26]. Fitting yields $A_1 = 1.831$, $A_2 = 0.196$, $B = 2.124$, $E = 5.533$, $\alpha_1 = \beta = 0.447$, $\alpha_2 = 0.671$. The detailed fitting process is written in §A.7.4.

**Usage** After fitting the parameters in Equation 8, the optimal vocabulary size can be obtained by finding the lowest loss w.r.t the vocabulary size, with a constraint of FLOPs budget. For example,

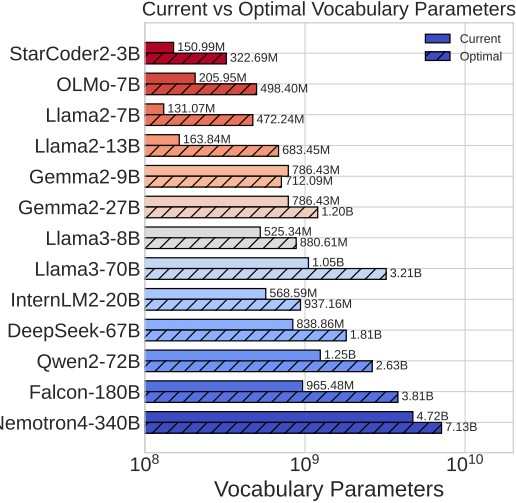

Figure 6: Vocabulary parameters of popular LLMs and predicted optimal vocabulary parameters at **their reported number of training tokens**, as determined by our Approach 3 (§4.3). Here we consider the practical scenarios where parameters and training data are not necessarily equally scaled. As illustrated, the vocabulary parameters remain predominantly underestimated. With the exception of Gemma2-9B, all models allocate a smaller vocabulary parameter count than our prediction.

given $N_{\mathrm{nv}}$ and FLOPs budget $C$, by replacing $[Hf(V)]$ with $C/(6(N_{\mathrm{nv}} + N_v))$ and finding the solution of $\frac{\partial \mathcal{L}_u}{\partial V} = 0$ via numerical search, we can get the prediction. The details of $\frac{\partial \mathcal{L}_u}{\partial V}$ is written in §A.2. Note that all of the proposed approaches can be used in optimally allocating $(N_{\mathrm{nv}}, N_{\mathrm{v}}, H)$ altogether, while Approach 3 is more flexible in predicting the locally optimal $N_{\mathrm{v}}$ when $(N_{\mathrm{nv}}, H)$ are not following the Chinchilla's law [26], *i.e.* equally-scaled law. The reason is that the loss formula in Approach 3 does not only considers the combinations $(N_{\mathrm{nv}}, N_{\mathrm{v}}, H)$ which reach the optimal given a certain training budget. By fixing $N_{\mathrm{nv}}$ and varying $C$ in Approach 3, we can predict the locally optimal vocabulary size with different amount of training characters. This property makes Approach 3 more valuable, since modern LLMs [70, 67, 3, 4, 7] usually leverage overly sufficient training data to build powerful models with relatively low inference costs.

In Figure 6, we remove the assumption [26] for the practical reason that the parameters and training data are not equally scaled. Then, we predict the locally optimal vocabulary parameters. It can be observed that the allocation of vocabulary parameters are typically under-estimated.

## 5  Discussion

**Predicting allocations for larger models**   Table 1 reports the predicted optimal vocabulary parameters and sizes based on the proposed three approaches, where the amount of training data is optimally

Table 1: We report the predicted optimal vocabulary parameters $N_v$ and the vocabulary size $V$ by the proposed three approaches given $N_{nv}$. We assume the training FLOPs are optimally allocated i.e. that the non-vocabulary parameters and training data are scaled equally. "App" denotes the approach.

| $N_{\mathrm{nv}}$ | $N_{\mathrm{v}}^{\mathrm{opt}}$-**App1** | $N_{\mathrm{v}}^{\mathrm{opt}}$-**App2** | $N_{\mathrm{v}}^{\mathrm{opt}}$-**App3** | **Dim.** | $V^{\mathrm{opt}}$-**App1** | $V^{\mathrm{opt}}$-**App2** | $V^{\mathrm{opt}}$-**App3** | **FLOPs Budget** |
|---|---|---|---|---|---|---|---|---|
| 3B | 0.1B | 0.1B | 0.1B | 3200 | 39K | 43K | 37K | $1.3e21$ |
| 7B | 0.3B | 0.3B | 0.2B | 4096 | 62K | 67K | 60K | $7.1e21$ |
| 13B | 0.4B | 0.5B | 0.4B | 5120 | 83K | 91K | 81K | $2.4e22$ |
| 30B | 0.9B | 0.9B | 0.9B | 6048 | 142K | 154K | 142K | $1.3e23$ |
| 70B | 1.7B | 1.9B | 1.8B | 8192 | 212K | 231K | 218K | $7.1e23$ |
| 130B | 2.9B | 3.2B | 3.0B | 12888 | 237K | 258K | 248K | $2.4e24$ |
| 300B | 5.8B | 6.4B | 6.3B | 16384 | 356K | 389K | 383K | $1.3e25$ |

Table 2: Zero-shot performance of models with $N_{nv} = 2.87B$ comparing the commonly used $V = 32K$ with our predicted optimal vocabulary $V^{opt}$. We consider the scenario where the number of training data is equally scaled with the non-vocabulary parameters. We report accuracy and standard deviation in percentages. Accuracy is normalized: The predicted likelihoods are divided by the length of each choice for multiple choices to eliminate the effect of text length on predictions.

| | $N_{\rm v}$ | $D$ | $H$ | ARC-C | ARC-E | Hellaswag | OBQA | WG | PIQA | BoolQ | Average |
|---|---|---|---|---|---|---|---|---|---|---|---|
| | | | | *FLOPs Budget 1.2e21 (Optimally-Allocated Training Data)* | | | | | | | |
| $V$=32K | 0.10B | 67.3B | 266.6B | $28.5_{\pm1.3}$ | $49.2_{\pm1.0}$ | $47.5_{\pm0.5}$ | $31.6_{\pm2.1}$ | $50.4_{\pm1.4}$ | $71.4_{\pm1.1}$ | $56.4_{\pm0.9}$ | 47.9 |
| $V^{opt}$=35K | 0.11B | 67.1B | 268.2B | $\mathbf{29.1}_{\pm1.3}$ | $\mathbf{50.6}_{\pm1.0}$ | $\mathbf{48.1}_{\pm0.5}$ | $31.6_{\pm2.1}$ | $\mathbf{51.9}_{\pm1.4}$ | $71.4_{\pm1.1}$ | $\mathbf{57.1}_{\pm0.9}$ | **48.5** |

Table 3: Zero-shot performance of models with $N_{\rm nv} = 2.87B$ comparing the commonly used $V = 32K$ with our predicted optimal vocabulary $V^{\rm opt}$ when **undertraining** or **overtraining**.

| | $N_{\rm v}$ | $D$ | $H$ | ARC-C | ARC-E | Hellaswag | OBQA | WG | PIQA | BoolQ | Average |
|---|---|---|---|---|---|---|---|---|---|---|---|
| | | | | *FLOPs Budget 2.8e20 (Insufficient Training Data, "Undertraining")* | | | | | | | |
| $V$=32K | 0.10B | 15.7B | 62.2B | $23.6_{\pm1.2}$ | $40.8_{\pm1.0}$ | $34.4_{\pm0.5}$ | $\mathbf{29.0}_{\pm2.0}$ | $49.7_{\pm1.4}$ | $64.9_{\pm1.1}$ | $59.8_{\pm0.9}$ | 43.2 |
| $V^{\rm opt}$=24K | 0.08B | 15.8B | 60.8B | $\mathbf{24.2}_{\pm1.3}$ | $\mathbf{42.2}_{\pm1.0}$ | $\mathbf{36.0}_{\pm0.5}$ | $28.6_{\pm2.0}$ | $\mathbf{50.0}_{\pm1.4}$ | $64.9_{\pm1.1}$ | $\mathbf{61.5}_{\pm0.9}$ | **43.9** |
| | | | | *FLOPs Budget 2.3e21 (Overly Sufficient Training Data, "Overtraining")* | | | | | | | |
| $V$=32K | 0.10B | 128.5B | 509.1B | $29.1_{\pm1.3}$ | $53.5_{\pm1.0}$ | $53.0_{\pm0.5}$ | $33.0_{\pm2.1}$ | $52.0_{\pm1.4}$ | $72.0_{\pm1.1}$ | $59.5_{\pm0.9}$ | 50.3 |
| $V^{\rm opt}$=43K | 0.14B | 127.0B | 517.5B | $\mathbf{32.0}_{\pm1.4}$ | $\mathbf{54.7}_{\pm1.0}$ | $\mathbf{54.1}_{\pm0.5}$ | $33.0_{\pm2.1}$ | $\mathbf{52.8}_{\pm1.4}$ | $\mathbf{72.6}_{\pm1.0}$ | $\mathbf{61.9}_{\pm0.9}$ | **51.6** |

allocated, *i.e.* equally scaled with the non-vocabulary parameters [26]. Aligned with the trend shown in Figure 1, *the predictions from all proposed approaches align closely*. $N_{\rm nv}$ should be scaled faster than $N_{\rm v}$. Notably, mainstream LLMs typically assign fewer parameters to vocabulary than what is optimal. However, the community is starting to shift to larger vocabularies, such as with Llama3 [41] having a 128K vocabulary size up from 32K of Llama2 [70]. However, scaling data is still the most critical part, and solving data scarcity issues should be a focus of future work [72].

To empirically verify our prediction, we train models with $N_{\rm nv} = 2.87B$ under a compute-optimal training FLOPs budget and evaluate them using lighteval [2]. For the baseline model we use the common vocabulary size of $V = 32K$. The other model uses $V^{\rm opt}$ as predicted by Approach 3. In Table 2, we show that the model allocated according to our vocabulary predictions yields better performance across multiple downstream tasks. This verifies that our predictions hold at scale.

**Experiments with scarce and excessive training data** Our prior experiments focus on the setting where training compute budget is the main constraint and we seek to allocate it optimally to parameters and training data. This is the typical setting in scaling law studies [30, 26, 52]. However, in the real world, we often deal with scarce data ("data-constrained [44]") forcing us to train sub-optimally or would like to make use of excessive data to train a smaller model that is cheaper to use [84]. To verify that our Approach 3 can handle these practical scenarios, we compare the model with $V = 32K$ and the model with the vocabulary size $V^{\rm opt}$ predicted by Approach 3. As shown in Table 3, our prediction enables a better model by only adjusting the vocabulary size in different FLOPs budgets.

In Figure 7, we further study the trend about how does the optimal vocabulary size shift with different number of training data. We only vary the amount of data but keep the non-vocabulary parameters fixed. The choices of vocabulary size are $8K$, $10K$, $16K$, $24K$, $32K$ and $48K$. Taking $N_{nv} = 302M$ as an example, **when available data is the bottleneck, the optimal vocabulary size decreases empirically**, *i.e.* $16K \rightarrow 10K$. This is a mechanism to prevent over-fitting. Conversely, when training on excessive amounts of data, *e.g.*, Llama3-8B uses much more training tokens than what would be compute-optimal for its budget, **the optimal vocabulary size increases**, *i.e.* $16K \rightarrow 24K$. Note that here we focus solely on training compute-optimal. It is also important to note that expanding the vocabulary size also increases the computational demands during inference. Therefore, we recommend **using the optimal vocabulary size corresponding to a given $N_{\rm nv}$, assuming optimal allocation of training data**, even in scenarios where overtraining may occur.

---

[2] https://github.com/huggingface/lighteval

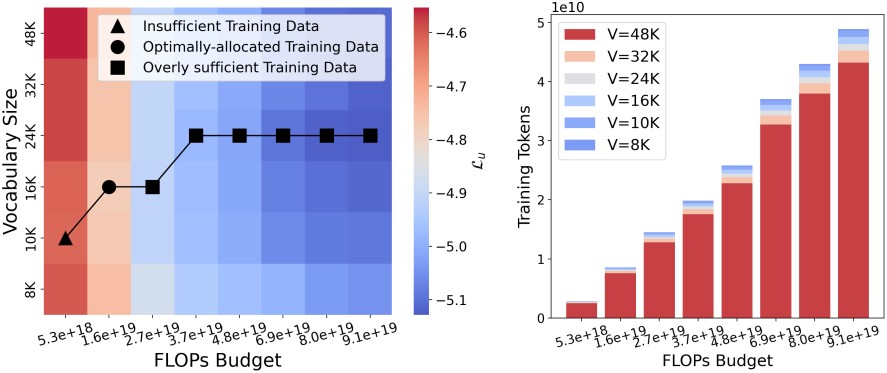

Figure 7: **Left:** The heatmap illustrates how the best vocabulary size among all choices of vocabularies shifts with the training data. The non-vocabulary parameter is fixed ($N_{nv} = 302M$). Each cell in the heatmap represents the loss given a certain FLOPs budget for a fair evaluation, with the color intensity indicating the loss value. The black line with markers denotes the best vocabulary size for each FLOPs budget, which basically increases as the number of training data increases. **Right:** The number of training tokens are slightly varying for different vocabulary sizes given a certain FLOPs budget. To keep FLOPs consistent, models with larger vocabulary sizes are trained on fewer tokens.

# 6    Related work

**Language models**    The Transformer [71] has proven to be a scalable architecture for language models, especially large language model (LLMs) [11, 14, 52, 47, 20, 29, 53, 70, 73, 41, 8, 4, 38, 25, 62, 67, 7, 39, 32, 88]. These models typically acquire a deep understanding of language enabling them to perform multiple tasks after a pre-training period and an optional fine-tuning period. Their capabilities include code generation [33, 3, 43, 87, 86], mathematical reasoning [76, 5], question answering [48, 45] among others. Given the expensive deployment costs required by the language models, various techniques can be adopted for efficient inference [64, 65, 78, 74, 36]. In our work, we pre-train large language models from scratch on English corpora and focus on their validation loss and downstream performance after training.

**Scaling laws**    Scaling laws aim to develop a predictive framework to find the best allocation of compute resources to maximize model performance. Besides language models, they have been studied in other domains[40, 68, 13]. For language models, Kaplan et al. [30] show that performance improves as a power law with more compute allocated to both parameters or data. Hoffmann et al. [26] show that the compute allocation of parameters and data should be scaled equally. Other work considers various cases such as downstream performance [23, 28, 55], inference time [57] or data constraints [44, 80]. However, the effect of vocabulary size has generally been ignored previously.

# 7    Conclusion

We investigate the impact of the vocabulary size in language models. We analyze and verify that there exists an optimal vocabulary size for a given FLOPs budget. Subsequently, we develop 3 approaches to predict the optimal vocabulary size. Our first approach uses empirical training runs across different IsoFLOPs regimes to fit a scaling law. The second approach investigates the FLOPs w.r.t. the vocabulary size and estimates the vocabulary size with derivatives. The third approach consists of a parametric function to predict the impact of different attributes on loss. Across all approaches, we find that while vocabulary parameters should be scaled slower than other parameters, they are still critical for performance and we can accurately predict their optimal allocation. We make predictions for larger models and empirically verify our approaches on up to 3B parameters and on varying amounts of training data. We show that models trained with an optimal vocabulary size as predicted by our approaches outperform models with a conventional vocabulary size under the same FLOPs budget.

## Acknowledgements

This work was supported by in part by the Theme-based Research Scheme (TRS) project T45-701/22-R of the Research Grants Council (RGC), Hong Kong SAR.

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

# A  Appendix

## A.1  The derivation of FLOPs w.r.t the vocabulary size for the Approach 2

Here we provide the detailed process of how we compute the extreme point of FLOPs $C$ with respect to $V$. From Kaplan et al. [30], we know that:

$$C \approx 6ND \approx 6(N_{\mathrm{nv}} + Vd)Hf(V). \tag{9}$$

We then compute the derivative $\frac{\partial C}{\partial V}$ as follows:

$$\begin{aligned}
\frac{\partial C}{\partial V} &= \frac{\partial}{\partial V}\left[6(N_{\mathrm{nv}} + dV)H\left(f(V)\right)\right] \\
&= \frac{\partial}{\partial V}\left[6(N_{\mathrm{nv}} + dV)H\left(a(\log(V))^2 + b\log(V) + c\right)\right] \\
&= 6H\left[(N_{\mathrm{nv}} + dV)\frac{d}{dV}\left(a(\log(V))^2 + b\log(V) + c\right)\right. \\
&\qquad\qquad \left. + \left(a(\log(V))^2 + b\log(V) + c\right)\frac{d}{dV}(N_{\mathrm{nv}} + dV)\right] \\
&= 6H\left[(N_{\mathrm{nv}} + Vd)\frac{2a\log(V) + b}{V} + \left(a(\log(V))^2 + b\log(V) + c\right)d\right].
\end{aligned}$$

The solution of $\frac{\partial C}{\partial V} = 0$ corresponds to the minimum point of the FLOPs. Since the variable $V$ in this equation is not separated conveniently, we use a numerical search method, specifically `scipy.optimize.fsolve`, to find the solution.

**Example demonstration**   Figure 8 illustrates the relationship between the derivative of FLOPs with respect to the vocabulary size $V$ and $V$ itself. Setting $V$ as the solution to $\frac{\partial C}{\partial V} = 0$, we find the point at which FLOPs are minimized. As depicted in Figure 8 (right), the FLOPs budget is fixed, and we observe how the training character varies with $V$. Notably, at the optimal vocabulary size $V$, the model expends the maximum number of training characters for a given budget. This observation provides insight into why an optimal vocabulary size exists for a given FLOPs budget.

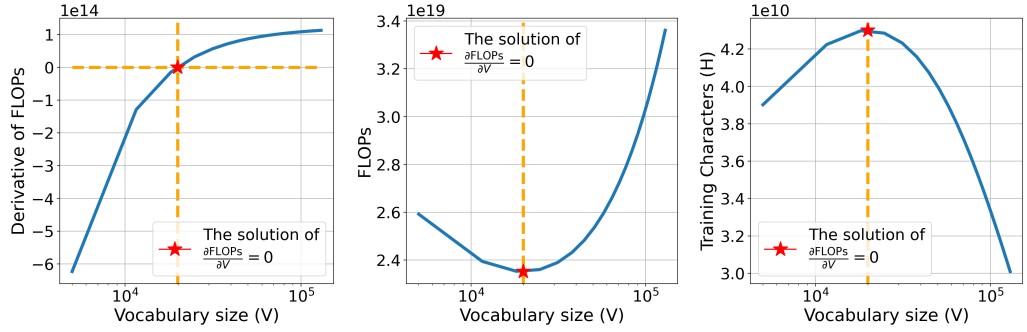

Figure 8: **Left:** The curve of the derivative of FLOPs with respect to vocabulary size $V$. The curve of $\frac{\partial C}{\partial V}$ increases as $V$ increases, and the FLOPs reach a minima at the solution of $\frac{\partial C}{\partial V} = 0$. **Middle:** The curve of FLOPs with respect to vocabulary size $V$, where $V$ reaches its optimal point $V$. **Right:** The curve of training characters with a given FLOPs budget. Take $N_{\mathrm{nv}} = 302M$ and $H = 43B$ as an example. The FLOPs budget is decided by the $N_{\mathrm{nv}}$, $H$ and the predicted $V$.

## A.2  The derivation of loss w.r.t the vocabulary size in Approach 3

Here we provide how we derive the loss w.r.t the vocabulary size given a FLOPs budget $C$ in Approach 3. After substituting the $[Hf(V)]$ with the $C/(6(N_{\mathrm{nv}} + N_v)$ based on Equation 9:

$$\mathcal{L}_u = -E + \frac{A_1}{N_{\mathrm{nv}}^{\alpha_1}} + \frac{A_2}{N_v^{\alpha_2}} + \frac{B}{[C/(6(N_{\mathrm{nv}} + N_v)]^\beta}. \tag{10}$$

The loss is solely dependent on the $N_v = Vd$, given a $N_{\mathrm{nv}}$. The derivative w.r.t. $V$ is:

$$\frac{\partial \mathcal{L}_u}{\partial V} = \frac{\partial}{\partial V}\left(\frac{A_2}{(Vd)^{\alpha_2}}\right) + \frac{\partial}{\partial V}\left(\frac{B}{\left(\frac{C}{6(N_{\mathrm{nv}}+Vd)}\right)^\beta}\right)$$

$$= -\alpha_2 \frac{A_2 d}{(Vd)^{\alpha_2+1}} + \beta \frac{B \frac{Cd}{6(N_{\mathrm{nv}}+Vd)^2}}{\left(\frac{F}{6(N_{\mathrm{nv}}+Vd)}\right)^{\beta+1}}.$$

The solution of $\frac{\partial \mathcal{L}_u}{\partial V} = 0$ corresponds to the optimal $V$. Similar with Approach 2, we use `scipy.optimize.fsolve` to find the solution.

### A.3 More visualizations for the analyses: Why the optimal vocabulary size is bounded by the compute

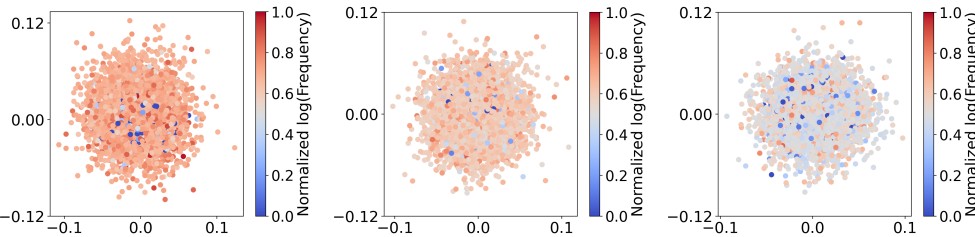

Figure 9: The SVD plots of the learned word embedding for V=4$K$ (left), V=16$K$ (middle) and V=64$K$ (right) for a model with $N_{\mathrm{nv}} = 85M$. Different colors represent different log frequencies.

**Word embeddings in a large vocabulary are hard to learn when FLOPs are constrained**
Previous studies have shown embeddings suffer from representation degradation, where low-frequency word embeddings cluster together due to limited parameter updating [24]. In Figure 9, we visualize how the word embeddings distribute using different vocabulary sizes. We use the average Euclidean distance among all the embeddings, $\mathcal{D}_{avg}$, to quantify the degree of clustering, which is 1.067, 1.011, and 0.952 for $V = 4K$, $V = 16K$ and $V = 64K$, respectively. Larger vocabularies (64$K$) lead to more clustering of embeddings, especially for infrequent words. This clustering suggests that they have been insufficiently trained. Conversely, a small-sized vocabulary (4$K$) and middle-sized vocabulary (16$K$) display a more dispersed distribution of embeddings. These observations suggest that there exists an optimal vocabulary size that balances lexicon coverage and sufficient updating of word embedding. Language models with large vocabulary sizes may have better lexicon coverage, but on the other hand, hinder the model's ability to sufficiently update the word embeddings, especially for low-frequency words.

### A.4 Exploration of Larger Range of Vocabulary Sizes

Because of computational resource constraints, the vocabulary sizes we used to fit the scaling laws are in the range of 4K to 96K. This range is sufficient to fit, because the optimal vocabulary sizes for all the training configurations we used fall in this range.

To further verify that there is always an optimal vocabulary size holds for a larger range of vocabulary lists, we increase the range of vocabulary sizes from 0.5K to 512K, with the $N_{\mathrm{nv}}$ fixed as 33M. As depicted in the Figure 10, the model's performance declines consistently as the vocabulary size increases beyond the optimal configuration. This figure shows loss curves for vocabulary sizes up to 512K, given a specific FLOPs budget. The data indicates a consistent degradation in model performance with the vocabulary size away from the optimal one. It suggests that there is a critical

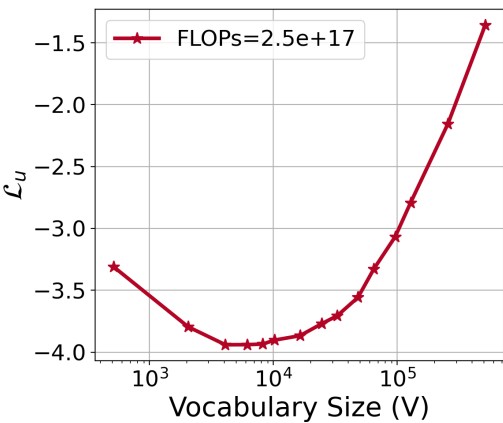

Figure 10: Loss curves with larger range of vocabulary sizes (from [4K, 96K] to [0.5K, 512K]), given a certain FLOPs budget. The model performance degrades consistently when the vocabulary size goes beyond the optimal configuration.

point beyond which the model's efficiency in handling the vocabulary diminishes. This exploration underscores the importance of carefully selecting the vocabulary size to maintain optimal model performance within the constraints of a given computational budget.

## A.5 The Vocabulary-insensitive Metric: $\mathcal{L}_u$ and BPC

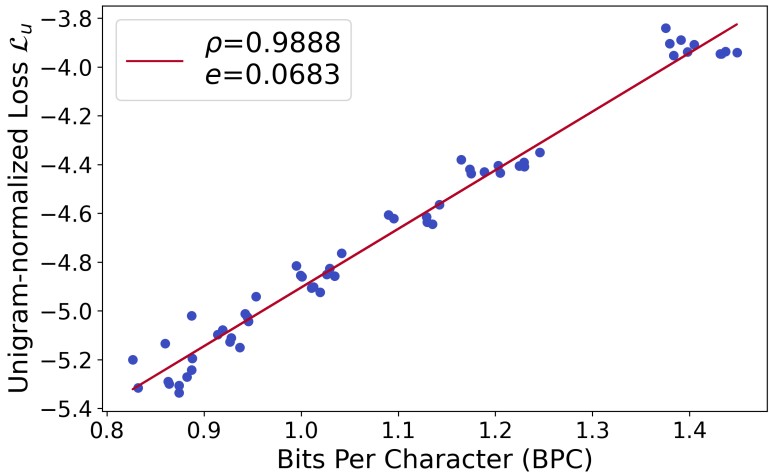

Figure 11: Correlation between the unigram-normalized loss $\mathcal{L}_u$ and BPC, where $\rho$ and $e$ denote the Pearson correlation coefficient and the root mean square error of the linear fit, respectively.

BPC reflects the ability to compress external text corpora [27], while the unigram-normalized loss reflects the model's ability to predict tokens normalized by the token frequency. Figure 11 shows the relationship between the Unigram-normalized Loss $\mathcal{L}_u$ and Bits Per Character (BPC) with a linear fit. We select the models of the final training steps for each $N_{nv}$ and each vocabulary size. The high correlation coefficient ($\rho = 0.9888$) and low error ($e = 0.0683$) indicate a strong linear relationship between these two metrics generally. However, it exists slight different trends due to different normalizations.

### A.6 More Explanations about Why We Separate Vocabulary Parameters from the total Model Parameters

Traditionally, scaling up model parameters in language models has been approached in two ways: increasing depth (i.e., the number of layers) or width (i.e., the hidden size). While extensive research has been conducted on these methods, current empirical practices often involve expanding both simultaneously [66]. This approach, however, may overlook crucial distinctions in how different parameters benefit from these expansions.

Non-vocabulary parameters can benefit from increases in both depth and width, allowing for more complex hierarchical representations and broader feature capture. In contrast, vocabulary parameters, associated with word embeddings and language model heads, are generally confined to a single layer, limiting their ability to benefit from increases in the model depth. Their primary avenue for expansion is through increasing the width. This disparity in growth potential between non-vocabulary and vocabulary parameters suggests that to maintain a balanced growth rate, it may be necessary to expand the vocabulary size along with the depth. This would allow the vocabulary parameters to keep pace with the growth of non-vocabulary parameters.

### A.7 Implementation details

#### A.7.1 Setting of model architecture, vocabulary size and training characters

We list the architectures of the models and the corresponding number of training characters in Table 4. For each model family, we fix the non-vocabulary parameters $N_{\mathrm{nv}}$ and vary the vocabulary size. We adopt the Llama architecture [69], except for the vocabulary size. For the vocabulary size, we use numbers divisible by 128 for compatibility with NVIDIA's tensor core to accelerate matrix multiplication [3]. Specifically, the vocabulary sizes we adopt for each model family are 4096, 6144, 8192, 10240, 16384, 24576, 32768, 48128, 64512 and 96256. The expected number of training tokens $D$ and characters $H$ vary slightly given a fixed number of non-vocabulary parameters and a FLOP budget. We use the middle-sized $V$ of 16384 to determine the number of training characters and the corresponding FLOPs budget, except for $N_{\mathrm{nv}} = 2870M$ we use $V = 32K$.

Table 4: The architectures of the models and the corresponding number of training characters adopted in our experiments.

| $N_{\mathrm{nv}}$ (M) | #Sequence Length | #Layers | #Heads | #Embedding Dim. | #Intermediate Size | Training Characters (B) |
|---|---|---|---|---|---|---|
| 33 | 2048 | 8 | 8 | 512 | 2048 | 4.3 |
| 85 | 2048 | 12 | 12 | 768 | 2048 | 11.1 |
| 151 | 2048 | 16 | 12 | 768 | 3072 | 19.6 |
| 302 | 2048 | 18 | 16 | 1024 | 4096 | 43.0 |
| 631 | 2048 | 20 | 24 | 1536 | 4800 | 101.6 |
| 1130 | 2048 | 22 | 32 | 2048 | 5632 | 201.3 |
| 2870 | 2048 | 24 | 32 | 3200 | 8192 | 509.3 |

#### A.7.2 The relationship between non-vocabulary parameters and embedding dimension

According to the observation in Kaplan et al. [30], the depth-width ratio has a relatively small effect on performance given the total non-vocabulary parameters. Thus, to ease the modeling of our scaling laws taking vocabulary size into account, we take the width (i.e. embedding dimension) as given following prior work [30, 26, 44, 70, 84]. The relationship between the non-vocabulary parameters $N_{\mathrm{nv}}$ and embedding dimension $d$ used in our experiments are in Table 5.

#### A.7.3 Training details

The maximum learning rate is set to 4e-4 and decays to 10% i.e. 4e-5 similar to prior scaling work [26, 44]. We use AdamW [37] as our optimizer and accelerate training with bfloat16 mixed

---

[3]https://docs.nvidia.com/deeplearning/performance/dl-performance-matrix-multiplication/index.html

Table 5: The relationship between the non-vocabulary parameters $N_{\mathrm{nv}}$ and the embedding dimension used in our experiments.

| Non-vocabulary Parameters $N_{\mathrm{nv}}$ | #Embedding Dim. |
|---|---|
| $N_{\mathrm{nv}} \le 50M$ | 512 |
| $50M < N_{\mathrm{nv}} \le 200M$ | 768 |
| $200M < N_{\mathrm{nv}} \le 500M$ | 1024 |
| $500M < N_{\mathrm{nv}} \le 1B$ | 1536 |
| $1B < N_{\mathrm{nv}} \le 2B$ | 2048 |
| $2B < N_{\mathrm{nv}} \le 5B$ | 3200 |
| $5B < N_{\mathrm{nv}} \le 10B$ | 4096 |
| $10B < N_{\mathrm{nv}} \le 20B$ | 5120 |
| $20B < N_{\mathrm{nv}} \le 50B$ | 6048 |
| $50B < N_{\mathrm{nv}} \le 100B$ | 8192 |
| $100B < N_{\mathrm{nv}} \le 200B$ | 12288 |
| $200B < N_{\mathrm{nv}} \le 500B$ | 16384 |
| $500B < N_{\mathrm{nv}} \le 1000B$ | 20480 |

precision training. For models with $N_{\mathrm{nv}} < 1130M$, we use a single node with 8 GPUs for training. Otherwise, we adopt the Megatron-LM framework [60] for multi-node training with 8 GPUs on each node. For our experiments with $N_{\mathrm{nv}} = 2870M$, it takes about 120 hours to train on over $500B$ training characters with 64 total GPUs. We use a global batch size of 512 for all runs and run all experiments on 40GB Nvidia-A100 GPUs.

### A.7.4 Fitting techniques

**Approach 1** To avoid numerical underflow and overflow of the fitting parameters, we fit the data in a logarithmic form inspired by Hoffmann et al. [26]. Taking $N_{\mathrm{nv}}$ as an example, we learn the parameters $k_1, \alpha_1$ by minimizing:

$$\min_{K_1,\alpha_1} \mathrm{Huber}_\delta(K_1 + \alpha_1 \log(C), \log(N_{\mathrm{nv}})), \tag{11}$$

where $K_1 = \log(k_1)$ and $\mathrm{Huber}_\delta$ denotes the Huber loss with delta value $\delta$ ($\delta$ is 0.001 in our paper). We use the LBFGS algorithm to find the local minima of the function. The later Approach 2 and 3 use the same optimization algorithm. We initialize all attributes from the same uniform grid where $K \in [-20, 15]$ and $\alpha \in [0, 1]$ with 20 initial guesses respectively. The fitting takes less than half of one minute.

To cheaply obtain more experimental data points, we perform interpolation of $(N_{\mathrm{nv}}, N_v, H)$ triplets in the logarithmic scale and predict the validation loss based on real data points. Then, we compute the required FLOPs for each data point using Equation 5.

**Approach 2** By using different $N_{\mathrm{nv}}$ and obtaining the corresponding optimal $N_v$ based on Equation 7, we have a set of $\{(N_{nv_i}, N_{v_i})|i = 1, ..., n\}$. Denoting $\mathcal{D}^{nv_i} = N_{nv_i}/N_{nv_0}$ and $\mathcal{D}^{v_i} = N_{v_i}/N_{v_0}$, we learn the scaling proportion $\gamma$ by minimizing:

$$\min_{\gamma} \mathrm{Huber}_\delta(\gamma * \log(\mathcal{D}^{nv_i}), \log(\mathcal{D}^{v_i})), \tag{12}$$

The initial guess of $\gamma$ is uniformly sampled from $[0, 1]$.

**Approach 3** We recast the designed vocabulary-dependent loss formula here:

$$\mathcal{L}_u = -E + \frac{A_1}{N_{\mathrm{nv}}^{\alpha_1}} + \frac{A_2}{N_v^{\alpha_2}} + \frac{B}{[Hf(V)]^\beta}, \tag{13}$$

where $\beta = \alpha_1$. In practice, we try to minimize:

$$\min_{a_1,a_2,b,e,\alpha_1,\alpha_2} \mathrm{Huber}_\delta\big( -\exp(e) + \exp(a_1 - \alpha_1 * \log(N_{\mathrm{nv}})) + \exp(a_2 - \alpha_2 * \log(N_{\mathrm{v}}) $$
$$+ \exp(b - \beta * \log([Hf(V)])), \quad \mathcal{L}_u\big),$$

where $A_1 = \exp(a_1), A_2 = \exp(a_2), B = \exp(b), E = \exp(e)$. We initialize all attributes from the same uniform grid where $a_1 \in [0, 5]$, $a_2 \in [0, 5]$, $b \in [0, 5]$, $e \in [0, 2]$, $\alpha_1 \in [0, 1]$ and $\alpha_2 \in [0, 1]$ with 3 initial guesses respectively. Given the prior that the scaling factor is typically ranged between 0 and 1 [26], we add a constraint $0.1 < \alpha_1, \alpha_2 < 1$ during fitting. The fitting also takes less than half of one minute.

### A.8 Details of fitting tokens-character relationship function $f(V)$

We train 25 tokenizers with the following vocabulary sizes: 1024, 2048, 3072, 4096, 5120, 6144, 7168, 8192, 9216, 10240, 12288, 16384, 20480, 24576, 28672, 32768, 48128, 64512, 78848, 96256, 128000, 256000, 512000, 1024000. Then, we train the tokenizers on a uniformly sampled version of the Slimpajama dataset.

Later, we apply the trained tokenizers on the validation set of the Slimpajama dataset and collect the number of tokens $D$ for each tokenizer with vocabulary size $V$. We use `scipy.optimize.curve_fit` to fit the parameters $a, b, c$ in $f(V)$ (§2.2).

### A.9 Robustness of the tokens-characters relationship function $f(V)$

**Robustness to the type of tokenizers** Besides the widely adopted BPE tokenizer used in our experiment, we also consider the unigram tokenizer and the word-based tokenizer. We visualize their tokens-characters ratio and corresponding predictive function in Figure 12. We find that our proposed formula of $f(V)$ is a good predictor for the tokens-character ratio, regardless of which tokenizer is used. This verifies the effectiveness of our proposed formula. The tokenization fertility of the unigram tokenizer is close to that of the BPE tokenizer as seen in their similar y-axis values, since they both employ subword-based tokenization. Meanwhile, the tokenization fertility of word-based tokenization is poor, thus requiring more tokens on average to compress characters.

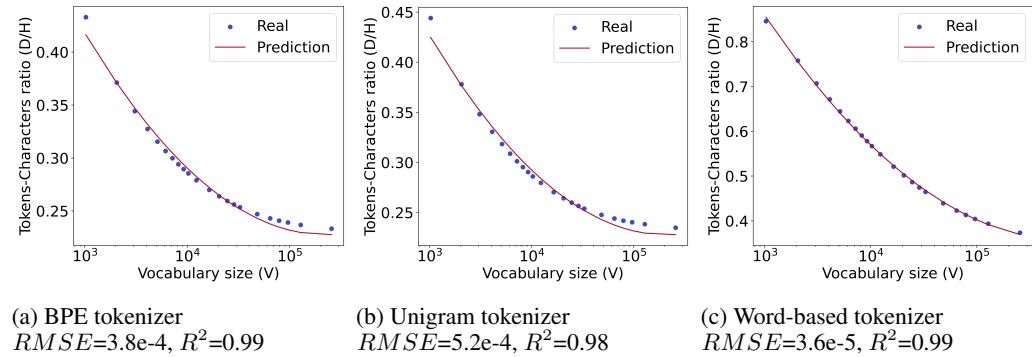

(a) BPE tokenizer
$RMSE$=3.8e-4, $R^2$=0.99

(b) Unigram tokenizer
$RMSE$=5.2e-4, $R^2$=0.98

(c) Word-based tokenizer
$RMSE$=3.6e-5, $R^2$=0.99

Figure 12: The modeling of function $f(V)$ with different tokenizers. RMSE and $R^2$ denote the relative mean square error and coefficient of determination, respectively.

**Robustness to the range of the vocabulary size** The quadratic function on the logarithmic value of vocabulary size that we propose can precisely predict the tokens-characters ratio with an RMSE of 1.5e-6 and $R^2$ of 0.99. However, as a quadratic function is single-peaked, increasing $V$ will increase the output value of $f(V) = a \log^2(V) + b \log V + c$ when $V$ is very large, *e.g.* $V > \exp(-b/2a) \approx 218K$ in our case.

Fortunately, when $V$ is sufficiently large, the tokenization fertility improvement of the tokenizer decays sharply, which results in almost no change to the value of $f(V)$. This is because the words in the training corpus can already be effectively covered by the vocabulary list when the vocabulary size is sufficiently large. In this extreme, the tokenization fertility of the corresponding tokenizer is approaching saturation, thus further increasing the vocabulary size will hardly improve the tokenization fertility.

As an example, there are about 2300M characters in the validation set of the Slimpajama corpus. A tokenizer using a vocabulary size of $2K$ would yield $140M$ fewer tokens than a $1K$ counterpart, but the number of tokens only decreases by $0.7M$ when going from a vocabulary size of $256K$ to

$257K$. Therefore, we add $min(V, 200K)$ before calculating $f(V)$ to ensure its decreasing nature. According to our prediction, a model with $300B$ parameters has an optimal vocabulary size of no more than $400K$ with a sufficient amount of training data. If we need to consider extremely large $V$ in the future, we can train tokenizers with larger $V$ in the process of fitting $f(V)$ to arrive at more precise predictions.

## A.10 Experimental verification on the fairness of the unigram-normalized language modeling loss

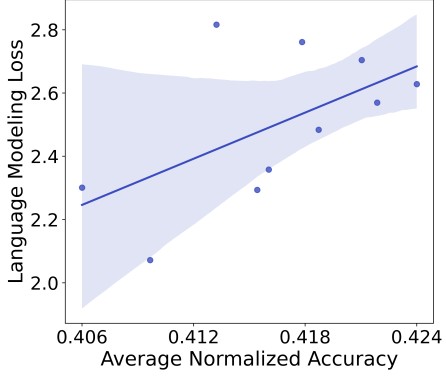 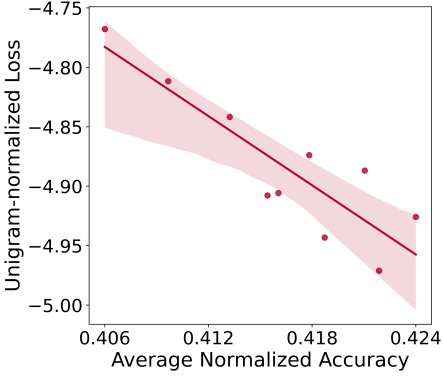

(a) Relationship between downstream task performance and the commonly-used language modeling loss.

(b) Relationship between downstream task performance and the unigram-normalized language modeling loss.

Figure 13: Empirical examination of the fairness of our unigram-normalized loss, $\mathcal{L}_u$. Dots correspond to trained models with varying vocabulary size. We plot their losses (y-axis) and performance on 7 downstream tasks (x-axis): WG [56], PIQA [9], OBQA [42], Hellaswag [83], BoolQ [16], ARC-E [17] and ARC-C [17]. The straight line reflects the results of the regression fit with the shade indicating the confidence interval.

In §2.2, we have explained that we use a unigram-normalized loss, $\mathcal{L}_u$, to fairly evaluate models that vary in vocabulary size. Here we empirically verify this choice. We train models with a fixed number of non-vocabulary parameters $N_{\mathrm{nv}}$ and embedding dimension $d$ but varying vocabulary sizes $V$. Thus, their vocabulary parameters $N_v$ also vary. We plot the final language model loss and unigram-normalized loss of these models compared to downstream performance in Figure 13. The language modeling loss exhibits a positive correlation with downstream performance: Models with a higher language modeling loss have better downstream performance. This is because our models with larger vocabularies naturally have a higher loss due to the objective function, yet they can be actually better models with better downstream performance. Our unigram-normalized loss solves this problem and exhibits the expected negative correlation between loss and downstream performance: a lower loss comes with better downstream performance. This empirically justifies our use of $\mathcal{L}_u$ throughout this work.

## A.11 Prediction for Llama3

While our primary experiments focus on the Llama2 vocabulary size, we also extend our conclusions to Llama3, predicting its optimal vocabulary under optimal compute allocation. As shown in Figure 14, we provide detailed predictions for various sizes of Llama3 models. Although Llama3 significantly increases its vocabulary size from 32K to 128K, our research suggests that this may still be insufficient for the larger model sizes of 70B and 400B.

## A.12 More Related Work

**Vocabulary in language models** The vocabulary of a language model influences its performance significantly [63, 75, 79]. A larger vocabulary size helps cover more words thus reducing the likelihood of out-of-vocabulary (OOV) cases [21]. Takahashi and Tanaka-Ishii [63] find that larger

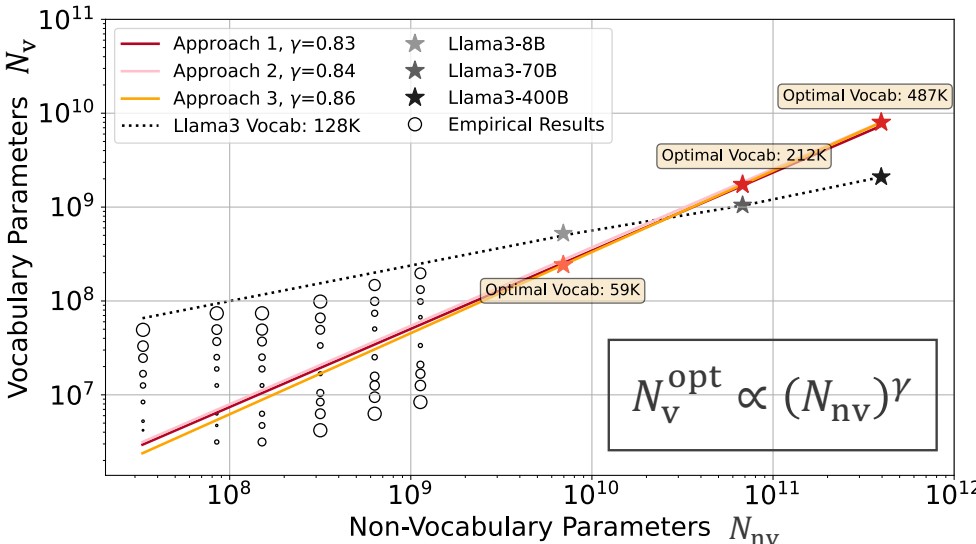

Figure 14: The replication of Figure 1 for Llama3. As shown, the predicted optimal vocabulary size for the Llama3-400B model is as high as 487K.

vocabularies are better at capturing the true statistical distribution of language. Similarly, expanding vocabulary in multilingual models [75, 15, 85, 34] improves performance, especially for low-resource languages. However, large vocabularies [31] increase the computational overhead during both training and generation phases. For example, Liao et al. [35] demonstrate that low-frequency words often lack sufficient examples to develop robust representations when vocabularies are excessively large. Dou et al. [19] reveal that expanding vocabularies during continual pre-training can lead to significant performance degradation for low-resource languages. More recently, Dagan et al. [18] explored the trade-offs associated with vocabulary size, proposing optimal vocabulary sizes for both memory efficiency and inference speed in code generation tasks. Our work complements these efforts by focusing on the broader impact of vocabulary size on downstream performance across various tasks. Specifically, we address a critical, under-explored question: How can we optimally allocate vocabulary size to maximize the downstream performance with the same compute budget?

**Byte-level language models** Recent work has explored byte-level language models [82, 77], which offer advantages in decoding efficiency and noise robustness compared to token-level models. However, typically limited to parameters under 1B, these models have not been effectively scaled up. Our scaling laws suggest that the limited vocabulary (i.e., 256 in byte-level language models) may constrain their performance, especially for larger models. The insight provides a potential explanation for the challenges in scaling byte-level models and implies that successful scaling of language models may require proportional increases in vocabulary size.

## B Limitation and future work

### B.1 Limitations of our proposed approaches

**Approach 1** The Approach 1 provides a broader solution by predicting the allocation of computational resources across non-vocabulary parameters, vocabulary parameters, and training data based on experimental data points. This method's strength lies in its holistic view, allowing for a balanced resource distribution that potentially enhances model efficiency and performance. However, this approach is constrained by the granularity and range of the experimental data points available, which can introduce errors in the fitting process. The requirement for substantial computational resources to perform these fittings may also limit its accessibility and scalability. Despite these challenges, when experimental data is ample and computational resources are sufficient, the Approach 1 can significantly refine the precision of resource allocation decisions in the development of large-scale language models.

**Approach 2**  By calculating the derivative of FLOPs with respect to the vocabulary size and solving for zero, this approach fundamentally relies on the precision of the FLOPs equation and our tokens-characters relationship function. Further, this method does not allow us to independently determine the optimal allocation of non-vocabulary parameters and training data size. Therefore, it necessitates information about the relationships between these attributes and the FLOPs budget from the experimentally fitted scaling laws, making this approach less useful in practice. Despite these limitations, the derivative-based approach offers notable advantages, including closely matched predictions with the scaling laws derived from actual experimental data in the Approach 2. Furthermore, its reliance on numerical solutions rather than exhaustive deep learning experiments makes it rapid and broadly applicable across various tokenizers, highlighting its utility in preliminary model configuration stages where quick estimates are key.

**Approach 3**  Similar with the Approach 1, the proposed Approach 3 requires multiple experimental runs across different non-vocabulary parameters, vocabulary sizes and number of training data. Therefore, the approach is constrained by the granularity and range of the experimental data points available to some extent. However, the proposed Approach 3 is flexible that it considers the fact that the non-vocabulary parameters and the number of training data are not always following the compute-optimal scaling laws [26], *i.e.*, equal scaling, in real-world applications.

## B.2  Larger models and different architectures

We have shown that our predictions hold for models with up to three billion parameters (§5). However, LLMs are often orders of magnitude larger, such as the 400-billion parameter Llama-3 model [41]. Further, we have decided to focus on dense transformer language models, as they are most commonly used for LLMs. However, many non-transformer models have been proposed and scaled up to billions of parameters [49, 50]. Exploring to what extent our findings hold in even larger models and with different architectures is a promising direction for future work.

## B.3  Parametric function for the loss when considering the vocabulary

Researchers [26, 44] consider modeling the language modeling loss with parametric functions in the form of $\mathcal{L} = P_1 + P_2/N^\alpha + P_3/D^\beta$, where $\{P_1, P_2, P_3, \alpha, \beta\}$ are learnable variables. The first term of loss represents the minimum achievable loss, and the second and third terms represent the contribution to the loss from the model size $N$ and number of training tokens $D$. The parametric function allows predicting the loss $L$ given $N$ and $D$ even if $(N,D)$ are not optimally allocated. In prior work, this loss formula accounts for changes in model size and training data but does not explicitly address the complexities introduced by varying vocabulary sizes. Incorporating vocabulary size into the loss predictor is challenging: Vocabulary size affects the model directly as well as the number of training tokens and the quality of tokenization by the tokenizer. A tokenizer with a large vocabulary size makes it easier to capture semantic information in raw text and reduces the frequency of out-of-vocabulary words. For instance, a large vocabulary size may cover common phrases, common subwords, and specialized terminology. Therefore, even if the same number of tokens are trained, the performance of the model trained on tokens with different qualities will be different.

Future work in this area could explore various parametric non-linear loss functions to predict the interactions between vocabulary size, model size, and training data with different compute allocations, not just the case of optimal compute allocation. Additionally, empirical studies on different datasets could help in understanding how vocabulary size impacts loss under varied data conditions, guiding the development of more adaptive loss prediction models.

## B.4  Extensions to multilingual and multimodal scenarios

Future work could extend the proposed approaches to encompass multilingual and multimodal scenarios. Multilingual models require a nuanced understanding of vocabulary due to linguistic diversity, which may affect the optimal vocabulary size and the computation of FLOPs differently across languages. Adapting these methods to consider linguistic features and tokenization variations could lead to more tailored and efficient resource allocations for multilingual models. Different languages compete with each other for the model's ability to allocate to that language [10], which

makes it necessary to take into account the relationship between different languages when setting the size of word lists for different languages in a multilingual scenario.

For multimodal models that integrate text with other data types such as images or video, the optimal vocabulary size might interact uniquely with non-linguistic parameters. Recent work [1, 68] models visual concepts in an autoregressive manner with tokenization like the processing of text data. It is interesting to explore the size of visual vocabulary size, *i.e.*, the codebook size [22], in the visual tasks and vision-language tasks. How to set the vocabulary size and the compute resource efficiently for different modalities remains an open issue.

## C  Potential social impact

The positive potential social impact of this research on vocabulary size in language model scaling is substantial. By optimizing large language models with the consideration of the vocabulary size and other attributes jointly, the paper provides a foundational understanding that can lead to more lightweight and cost-effective pre-trained large language models. This efficiency can democratize access to advanced language processing technologies, making it feasible for smaller organizations and the general public to benefit from powerful AI tools. Such advancements can benefit various domains, for example, improve accessibility features for individuals with disabilities, where efficient language models can be used to analyze medical records and assist in diagnostics. Furthermore, the reduction in computational requirements for training these models can lead to a decrease in energy usage, contributing positively to environmental sustainability efforts.

On the other hand, the misuse of pretrained language models may pose risks, including the creation of highly realistic deepfakes that can spread disinformation and undermine trust in media and institutions. These models can generate misleading content, automate cyberattacks through convincing phishing schemes, and produce large-scale spam, degrading online communication. Additionally, they can be used to generate harmful or abusive content, such as hate speech, which perpetuates discrimination and harms vulnerable populations. To mitigate these risks, it is crucial to develop trustworthy language models, implement robust monitoring systems, and foster collaboration among researchers, policymakers, and users.

