# OpenReview forum: "Scaling Laws with Vocabulary: Larger Models Deserve Larger Vocabularies"
_NeurIPS.cc/2024/Conference — NeurIPS 2024 poster_

### Official Review · Reviewer_zt86 · 2024-06-24

**Soundness:** 3
**Presentation:** 3
**Contribution:** 3
**Rating:** 6
**Confidence:** 4

**Summary:**

The paper presents empirical scaling laws for the size of the vocabulary for LLMs. The findings in the paper are:

- Empirically, the vocabulary size minimizing the loss increases when FLOPs are increased (Fig 2, right, Fig 3)
- Through mathematical derivations from scaling laws, the optimal vocabulary size decreases when the embedding size increases. (Fig 4).
- With a given FLOPs budget, 43k vocav size beats 32k on academic benchmarks like Boolq (Table 2.)

**Strengths:**

- Vocabulary scaling laws are an interesting and novel research direction.
- The paper is well written.

**Weaknesses:**

- The results are probably mostly applicable to a small number of well-funded labs.

**Questions:**

1. The abstract states that “beyond the conventional 32K” – is this really the convention? See e.g. GPT4o.
2. How does Table 2 look if you train on the same number of tokens instead of using the same FLOPs budget?

---

> ### Author Rebuttal · Authors · 2024-08-06
>
> ### W1: The results are probably mostly applicable to a small number of well-funded labs.
>
> Thanks for pointing this out! We want to clarify that we are not a well-funded lab either. Due to our limited computing resources, we can only afford to train models with up to 3B parameters in our experiments, as the cost of validating scaling law experiments is indeed very high.
>
> However, we believe that our conclusions are beneficial to the general research community, especially for small labs. Our scaling laws with vocabulary provides a compute-optimal allocation suggestion, enabling small labs to train high-performance models without repeatedly trying different vocabulary configurations, thereby saving computing resources.
>
> Even for teams who want to conduct scaling law experiments themselves, our derivative-based method offers a simple and feasible approach based on theoretical derivation. Researchers do not need to run a large number of scaling law experiments to obtain a good vocabulary configuration. This is particularly advantageous for small labs. We will also make all our scaling law experimental results public so that more people can benefit from our work.
>
> ### Q1: The abstract states that “beyond the conventional 32K” – is this really the convention? See e.g. GPT4o.
> Thank you for your insightful question! We acknowledge that there is no single "conventional" vocabulary size for language models, as it can vary based on the pre-training corpus and the intended use case. A vocabulary size of 32K is widely regarded as a common choice, particularly for models trained on English-centric corpora, such as Llama-1, Llama-2, and Mistral. Since our work primarily utilizes the English-centric SlimPajama corpus for pre-training, we have adopted the 32K vocabulary setting employed by these models as a "conventional" vocabulary size. We will modify the statement in the abstract accordingly to reflect this clarification.
>
> As for GPT4o, we think its vocabulary size is relatively larger because it is designed to handle multiple languages (e.g., Chinese). This also highlights an important consideration for future research: determining the optimal vocabulary size for multilingual models, which we have discussed in Appendix B.4.
>
> Our broader goal is to draw attention to the importance of vocabulary size in training language models and to encourage the appropriate allocation of computational resources for this aspect. Recently, there has been a shift in the industry, with major companies recognizing that their previous allocations for vocabulary were insufficient. For example, Llama has increased its vocabulary size from 32K to 128K, reflecting this evolving understanding. We hope this clarification helps, and will add the discussion in the revised version.
>
>
> ### Q2: How does Table 2 look if you train on the same number of tokens instead of using the same FLOPs budget?
>
> Thanks for your prompting question! As you suggested, we also trained the model using the same number of tokens, i.e., 129B tokens, beyond the same FLOPs budget setting. As shown in the following table, the performance of the model with the suggested vocabulary size of 43K improves further compared to the 32K vocabulary size when using the same number of training tokens. We will add the results in the revised version.
>
> | **$V$** | **$N_v$** | **$D$** | **Winogrande** | **PIQA** | **OBQA** | **Hellaswag** | **BoolQ** | **ARC-E** | **ARC-C** | **Average** |
> |---------|-----------|---------|----------------|----------|----------|---------------|-----------|-----------|-----------|-------------|
> | 32K (Baseline)     | 0.20B     | 129B    | 55.7 ± 1.4     | 72.6 ± 1.0 | **34.4** ± 2.1 | 55.1 ± 0.5  | 60.1 ± 0.9  | 53.4 ± 1.0  | 29.1 ± 1.3  | 51.5        |
> | 43K (Ours with same FLOPs)    | 0.27B     | 125B    | **58.7**±1.4 |**72.7**±1.0 |33.0±2.1 |55.7±0.5 |62.3±0.8  | 55.0±1.0  | 31.5±1.4 |  52.7
>  | 43K (Ours with same Tokens)    | 0.27B     | 129B    | 58.6±1.4 |  **72.7**±1.0  | 33.6±2.1  | **55.8**±0.5  | **62.4**±0.9  | **55.5**±1.0  | **31.5**±1.4 |  **52.9**

---

> ### Author Response · Authors · 2024-08-11
> **Any New Comments Would be Greatly Appreciated**
>
> Dear Reviewer zt86,
>
> We are deeply grateful for your detailed review and the insightful suggestions you provided for our paper. We have carefully considered and responded to each of your comments in our rebuttal.
>
> As the Author-Review Discussion period is coming to an end, we want to ensure that all your concerns have been thoroughly addressed. If there are any remaining questions or issues, we would be glad to offer further clarification or make any necessary revisions.Thank you once again for your valuable feedback.
>
> Best regards,
>
> The Authors

---

### Official Review · Reviewer_Y8zo · 2024-07-12

**Soundness:** 3
**Presentation:** 3
**Contribution:** 3
**Rating:** 6
**Confidence:** 4

**Summary:**

This study primarily explores the role of vocabulary size in scaling large language models (LLMs). Traditional research has focused on model parameters and training data size, often overlooking the impact of vocabulary size. While intuitively larger vocabularies can enable more efficient tokenization by representing sentences with fewer tokens, they also increase the risk of under-fitting representations for rare tokens. By training models ranging from 33M to 3B parameters on up to 510B characters with various vocabulary configurations, we discovered that the optimal vocabulary size is constrained by the computational budget. We propose two methods to determine the optimal vocabulary size: an empirical IsoFLOPs approach and a fast derivative-based approach. Both methods indicate that vocabulary parameters should be scaled slower than non-vocabulary parameters. Nonetheless, vocabulary parameters are critical for performance and are under-allocated in current LLMs. By increasing the vocabulary size beyond the conventional 32K, we trained a better 3B parameter model despite using fewer training tokens. Our work reveals the underestimated role of vocabulary and the necessity of jointly considering vocabulary size, model parameters, and training data for efficient scaling.

**Strengths:**

1. The study takes a holistic approach by examining the role of vocabulary size in the scaling of large language models (LLMs), addressing a gap in traditional research that often overlooks this aspect.
2. The introduction of two novel methods—an empirical IsoFLOPs approach and a fast derivative-based approach—for determining the optimal vocabulary size showcases the study's innovation and practical contributions.
3. Demonstrating that increasing vocabulary size beyond the conventional 32K can lead to better model performance with fewer training tokens highlights the practical implications and potential for efficiency gains.

**Weaknesses:**

1. Lacks performance on large-scale models, such as whether increasing the vocabulary size to a greater extent performs better than existing models in the market. Table 2's experiments look a little bit less.
2. IsoFLOPs method is very sensitive, the experiments also looks not enough.

**Questions:**

In Use Case section, the authors mentioned that  (2) One has already conducted scaling law experiments following the Chinchilla laws with a fixed vocabulary size (e.g., 244 32K) and aims to estimate the optimal vocabulary size for a given model parameter. In determining the scaling law for the relationship between non-embedding model size and data (such as the Chinchilla law), why is it assumed that the vocabulary size is independent of these two factors

**Limitations:**

Yes, the authors addressed limitations.

---

> ### Author Rebuttal · Authors · 2024-08-06
>
> ### W1: Lacks performance on large-scale models, such as whether increasing the vocabulary size to a greater extent performs better than existing models in the market. Table 2's experiments look a little bit less.
>
>
> Thank you for raising this concern. We share the intention to compete with existing powerful models in the market, such as Llama-2-7B. However, training a 7B model on 2 trillion tokens from scratch is far beyond our current computational resources. Nevertheless, we have evaluated our 3B models on more benchmarks to alleviate your concern.
>
> Specifically, we have added new experimental results on the following benchmarks:
> - **MMLU**:Massive Multitask Language Understanding benchmark for broad domain language evaluation.
> - **CommonsenseQA**: A multiple-choice QA dataset for measuring commonsense knowledge.
> - **CoQA**: Conversational question answering tasks to test dialog understanding.
> - **TruthfulQA**: A QA task aimed at evaluating the truthfulness and factual accuracy of model responses.
> - **Lambada**: Tasks designed to predict the endings of text passages, testing language prediction skills.
>
> The following table combines the original Table 2 in the paper and the new experimental results. As shown, our prediction enables better model performance by adjusting the vocabulary size within different FLOPs budgets. The 3B model with a 43K vocabulary size outperforms the 32K counterpart on 11 out of 12 tasks using the same FLOPs budget. For example, we improve performance on ARC-C from 29.1 to 31.5. In conclusion, the model using our suggested vocabulary size (i.e., 43K) consistently outperforms its counterpart (i.e., 32K) by a clear margin.
>
>
> | Tasks         | Metric              | $V$=32K (Baseline)   | $V^{opt}$=43K (Ours) |
> |---------------|---------------------|-----------|---------------|
> | Winogrande    | Normalized Accuracy | 55.7±1.4  | **58.7**±1.4      |
> | PIQA          | Normalized Accuracy | 72.6±1.0  | **72.7**±1.0      |
> | OBQA          | Normalized Accuracy | **34.4**±2.1  | 33.0±2.1      |
> | Hellaswag     | Normalized Accuracy | 55.1±0.5  | **55.7**±0.5      |
> | BoolQ         | Normalized Accuracy | 60.1±0.9  | **62.3**±0.8      |
> | ARC-E         | Normalized Accuracy | 53.4±1.0  | **55.0**±1.0      |
> | ARC-C         | Normalized Accuracy | 29.1±1.3  | **31.5**±1.4  |
> | MMLU          | Normalized Accuracy | 25.0±0.4  | **25.5**±0.4  |
> | CommonsenseQA | Normalized Accuracy | 20.2±1.2  | **21.0**±1.1  |
> | CoQA          | Exact Match         | 32.3± 2.0 | **37.4**± 2.0 |
> | TruthfulQA    | BLEU                | 30.4±1.6  | **31.3**±1.6  |
> | Lambada       | Normalized Accuracy | 43.0±0.7  | **44.9**±0.7  |
>
> Another feasible way to compete with models in the market would be continual pre-training with a larger vocabulary size. We believe this is a good topic for discussion, but it involves several non-trivial research challenges not strongly related to the main contributions of this paper, i.e., exploring the optimal compute allocation considering vocabulary sizes. Therefore, we will discuss it in the revised version and leave it as an important future work. The challenges we will discuss include:
> - Expanding the vocabulary necessitates changes in the tokenization process, which can lead to inconsistencies in token segmentation.
> - Ensuring that these new embeddings are compatible and effectively integrate with the pre-trained embeddings is non-trivial.
> - Catastrophic forgetting of old word embeddings when learning new word embeddings.
>
> We will discuss all the above in the revised version. Thank you again for your valuable comments.
>
>
>
> ### W2.1: IsoFLOPs method is very sensitive.
>
> Thank you for your insightful question. You raise a valid point – the IsoFLOPs-based approach can be sensitive to some extent, depending on the granularity, range, and quality of the fitting data. Since the pioneering work on scaling laws by Kaplan et al. 2020 [1] and Hoffmann et al. 2022 [2], the IsoFLOPs-based approach has become a widely-used tool to study the trend of model performance [3]. We have discussed it in our Appendix B.1, and we will add more details on how to reduce sensitivity, such as outlier data removal and repeated experiments, in our polished version.
>
> To evaluate the goodness of fit, we use relative mean square error (rMSE) and the coefficient of determination (R^2). As shown in the table below (also in Figure 3), the results indicate a good fit, with rMSE < 0.001 and $R^2$ >= 0.89 for all the considered attributes: non-vocabulary parameters ($N_{nv}$), vocabulary parameters ($N_v$), and training characters ($H$). This suggests that these attributes follow a power law with respect to the FLOPs budget.
>
> |  | $N_{nv}$ | $N_v$ | $H$ |
> |--------|----------|------|-----|
> | rMSE   | 0.00026  | 0.00051 | 0.00017 |
> | $R^2$  | 0.93  | 0.89  | 0.96  |
>
> Furthermore, the optimal vocabulary predictions (Table 1) from the IsoFLOPs-based method and the derivative-based method are aligned across small-scale and large-scale models. This independent verification by the derivative-based method validates the predictions from the IsoFLOPs-based method. Therefore, we believe that the IsoFLOPs-based method works well in our case.
>
> [1] Jared Kaplan, Sam McCandlish, Tom Henighan, Tom B Brown, Benjamin Chess, Rewon Child, Scott Gray, Alec Radford, Jeffrey Wu, and Dario Amodei. 2020. Scaling laws for neural language models. arXiv preprint arXiv:2001.08361.
>
> [2] Jordan Hoffmann, Sebastian Borgeaud, Arthur Mensch, Elena Buchatskaya, Trevor Cai, Eliza Rutherford, Diego de Las Casas, Lisa Anne Hendricks, Johannes Welbl, Aidan Clark, et al. 2022. Training compute-optimal large language models. arXiv preprint arXiv:2203.15556
>
> [3] Jack W Rae, Sebastian Borgeaud, Trevor Cai, Katie Millican, Jordan Hoffmann, Francis Song, John Aslanides, Sarah Henderson, Roman Ring, Susannah Young, et al. 2021. Scaling language models: Methods, analysis & insights from training gopher. arXiv preprint arXiv:2112.11446.

---

> > ### Comment · Reviewer_Y8zo · 2024-08-12
> >
> > Thank you for your thorough response to my questions. I am pleased to see the new results. However, there is room for improvement in this paper, such as Figure 3 (left), where the distribution of model sizes is very uneven. For example, for models below 100M, the basic points of model size are concentrated at 33M and ~90M, which will seriously affect the exponential factor of the fitted curve.

---

> > > ### Author Response · Authors · 2024-08-12
> > > **Response to reviewer Y8zo**
> > >
> > > Dear Reviewer Y8zo,
> > >
> > > Thank you for your timely feedback. We're pleased that you found our new results compelling and appreciate your insightful comments on areas for improvement, particularly regarding Figure 3 (left). We would like to offer the following clarifications:
> > >
> > > Our study expands on traditional scaling law approaches by incorporating additional parameters: Nnv (non-vocabulary parameters), Nv (vocabulary parameters), and H (training characters). This contrasts with previous work that primarily considered N (total model parameters) and D (training tokens). The inclusion of these additional factors adds complexity to the scaling law fitting process. Given the high computational costs associated with scaling law experiments, we concentrated our efforts on a limited set of non-vocabulary parameters while exploring a broader range of vocabulary parameters. This focus aligns with the primary objective of our research: investigating the impact of vocabulary size on model performance.
> > >
> > > To achieve this, we deliberately selected 10 groups with varying vocabulary sizes for each of the 6 fixed groups of non-vocabulary parameters. While this design choice accounts for the uneven distribution seen in Figure 3 (left), it also allows for a more thorough exploration of vocabulary size effects, as demonstrated by the diverse data points in Figure 3 (middle). Our experimental design and results robustly support our main conclusion: larger models benefit from larger vocabularies.
> > >
> > > We believe these clarifications provide a more comprehensive understanding of our research methodology and findings. We appreciate your careful review and hope these explanations enhance your confidence in our paper. We look forward to any further feedback you may have.
> > >
> > > Best regards,
> > >
> > > The Authors

---

> ### Author Response · Authors · 2024-08-06
> **Response to reviewer Y8zo's second part**
>
> ### W2.2: The experiments also looks not enough.
> It is noteworthy that we conduct extensive experiments on **1200 models pre-trained from scratch** (6 non-vocabulary parameters settings x 10 vocabulary sizes settings x 20 training data settings) for the fitting of our vocabulary scaling law. The key contributions in this paper is the several findings about how the vocabulary affects the model performance and how much compute should be allocated on the vocabulary based on the proposed 2 approaches.
>
> Following the previous study [1,2,3], we mainly use the held-out validation loss value for the evaluation of the trained 1200 models. It is a better metric than the downstream tasks performance as the held-out loss  provides an unbiased measure of the model’s ability to generalize to new data, but also enjoys high computing efficiency. Instead, the performance of downstream tasks has a great variety across different tasks, which is not suitable as the main evaluation metric.
>
> The evaluation of downstream tasks is part of the ways to verify our prediction, therefore we do not take too much content to discuss it in our main paper. For downstream tasks, we conduct more experiments in the answer of your #Q1. The new results will be added in our polished version.
>
> ### Q1: In determining the scaling law for the relationship between non-embedding model size and data (such as the Chinchilla law), why is it assumed that the vocabulary size is independent of these two factors.
>
> Thanks for your question! We do not assume that the vocabulary size is independent of parameters and data. Instead, we make some adjustments in the Section of Preliminary: 1) We break down the total parameters into non-vocabulary parameters and vocabulary parameters; 2) We measure data not in tokens but in training characters.
>
> By doing so, the vocabulary size $V$ is independent with the non-vocabulary parameters $N_{nv}$ and the number of training characters $H$. In an experimental configuration, the developers can vary the vocabulary size without affecting non-vocabulary parameters or training characters.
>
> Then, we details our motivation why we separate the vocabulary parameter and  non-vocabulary parameter below:
>
> Traditionally, scaling up model parameters in language models has been approached in two ways: increasing depth (i.e., the number of layers) or width (i.e., the hidden size). Current empirical practices often involve expanding both simultaneously [4]. This approach overlook crucial distinctions in how different parameters benefit from parameters expansions. Non-vocabulary parameters can benefit from increases in both depth and width, allowing for more complex hierarchical representations and broader feature capture. In contrast, vocabulary parameters, associated with word embeddings and language model heads, are generally confined to a single layer, limiting their ability to benefit from increases in the model depth. This disparity in growth potential between non-vocabulary and vocabulary parameters suggests that to maintain a balanced growth rate, it is better to separate the vocabulary parameter  and  non-vocabulary parameter into consideration.
>
>
> [4] Yi Tay, Mostafa Dehghani, Samira Abnar, Hyung Chung, William Fedus, Jinfeng Rao, Sharan Narang, Vinh Tran, Dani Yogatama, and Donald Metzler. 2023. Scaling Laws vs Model Architectures: How does Inductive Bias Influence Scaling? In Findings of the Association for Computational Linguistics: EMNLP 2023, pages 12342–12364, Singapore. Association for Computational Linguistic.

---

> ### Author Response · Authors · 2024-08-11
> **Any New Comments Would be Greatly Appreciated**
>
> Dear Reviewer Y8zo
>
> We sincerely appreciate your thorough review and the valuable suggestions and comments you provided for our paper. We have carefully considered each of your points and have addressed them in detail in our rebuttal.
>
> With the Author-Review Discussion period nearing its conclusion, we want to ensure that all your concerns have been fully addressed. If there are any questions or unresolved issues, we are eager to provide further clarification or make any necessary revisions.
>
> Thank you again for your thoughtful feedback.
>
> Best regards,
>
> The Authors

---

### Official Review · Reviewer_EsaU · 2024-07-18

**Soundness:** 3
**Presentation:** 3
**Contribution:** 3
**Rating:** 7
**Confidence:** 2

**Summary:**

This paper investigates the impact of vocabulary size on the efficiency of large language models (LLMs). Using models with 33 million to 3 billion parameters, it finds that optimal vocabulary size is limited by computational resources. The study introduces two methods to determine the best vocabulary size, showing that vocabulary parameters should scale slower than other parameters. Results highlight the significant yet underestimated role of vocabulary in scaling LLMs effectively, suggesting that larger vocabularies can improve model performance.

**Strengths:**

The paper addresses a unique aspect of language model scaling by investigating the impact of vocabulary size on model performance, a dimension that is often overlooked in LLM research.

The introduction of two novel methods to determine the optimal vocabulary size—empirical IsoFLOPs and a derivative-based approach—provides practical tools for optimizing LLM training and deployment.

The analysis in this paper is quite in-depth, and some conclusions can provide references for subsequent LLM training efforts.

**Weaknesses:**

This paper conducted experiments on language models of various parameter sizes, but the largest model tested was only 3 billion parameters. It would be better if we could further verify models with more than 7 billion parameters. I believe both the industrial and academic communities are eager to know whether the scaling law for vocabulary can generalize to larger models.

**Questions:**

NA

**Limitations:**

This paper discusses the limitations in Appendix B.

---

> ### Author Rebuttal · Authors · 2024-08-06
>
> ### W1: This paper conducted experiments on language models of various parameter sizes, but the largest model tested was only 3 billion parameters.
> We acknowledge the importance of evaluating our approach on larger models to establish its scalability. Increasing the model size necessitates pre-training on a larger corpus, which in turn demands more computational resources. For instance, conducting pre-training experiments on 7B models would require an immense computational budget exceeding 10^22 FLOPs, translating to approximately 6 weeks of training time on a cluster with 64 A100 GPUs. However, such a substantial level of computational resources is currently beyond our reach during the rebuttal period. Despite our desire to explore larger model sizes, we are constrained by the practical limitations of our available resources.
>
> Nonetheless, the significance of the scaling law lies in investigating it through experiments on a relatively small scale to help us reasonably allocate computational resources when training a large model, thus avoiding wasted computational power. Our experiments with 1200 pre-trained models have demonstrated the existence of an optimal vocabulary size under FLOPs constraints, and the predictions from the theoretical analysis (derivative-based approach) and experimental fitting (IsoFLOPs-based approach) agree with each other. In fact, we have also made predictions for the pre-training of a 300B model, and the two approaches align well, and so we believe it should work for larger models.
>
> Furthermore, the change in vocabulary size from 32K in Llama-2 to 128K in Llama-3, which resulted in performance improvement, can be also seen as a verification of our conclusion regarding increasing the vocabulary size when there are more computational budgets.
>
> In conclusion, we appreciate your suggestions and we will try our best to conduct more experiments on larger models. We hope you can understand our computational limitations. We will also discuss this in the revised version.

---

> ### Author Response · Authors · 2024-08-12
> **Any New Comments Would be Greatly Appreciated**
>
> Dear Reviewer EsaU,
>
> We are truly appreciative of the time and effort you have dedicated to reviewing our paper. Your thoughtful feedback and constructive suggestions are valuable to us. We have carefully addressed your comments in our rebuttal to enhance the quality of our work.  As we approach the final days of the Author-Review Discussion period, we would like to ensure that all your concerns have been comprehensively addressed. Should there be any remaining questions or issues, we are willing to provide further clarification or additional revisions.Thank you once again for your insightful contributions to our work.
>
> Best regards,
>
> The Authors

---

### Author Rebuttal · Authors · 2024-08-06

### General Response

We are grateful for the reviewers' efforts and the recognition of our contributions:
- **Novel Research Topic:** The paper explores the unique impact of vocabulary size on language model performance, an aspect often overlooked in LLM research [EsaU,Y8zo,zt86].
- **Analyses:** We provide in-depth analyses why it exists a  optimal vocabulary size with the FLOPs and the optimal vocabulary size increases with more FLOPs, with theoretical anayses in Appendix A.1 and demostration experiments in Sec 3 and Apppendix A.2. [EsaU,zt86]
- **Experiments:** The paper includes two effective methods (IsoFLOPs and a derivative-based approach) to predict the optimal vocabulary setting. Extensive experiments on **1200 pre-trained models** (6 non-vocabulary parameters settings x 10 vocabulary sizes settings x 20 number of training data settings) are conducted. [EsaU,zt86]
- **Applications:** The study's findings offer two practical tools for optimizing the compute allocaiton of LLMs training, with the consideration of the vocaulary size. [EsaU,Y8zo,zt86]

In response to the feedback of reviewers, we have performed additional analyses and experiments to address the raised concerns. Below, we summarize our responses and the improvements made to our paper:
- We evaluate the 3B pre-trained models with different vocabulary sizes on more downstream tasks. The model that uses our suggested vocabulary size outperforms its counterpart by a clear margin.
- We supplement the paper with a new perspective from parameter growing to demontrate that the vocabulary parameters need to be separately considered from the total parameters, and larger models deserve larger vocabularies.
- We compare the models with the same training tokens instead of the same training FLOPs as asked.

All of the suggestions will be considered in our polished version.

---

### Decision · Program_Chairs · 2024-09-25

**Decision:**

Accept (poster)

**Comment:**

This paper studies the important aspect of vocabulary size in the context of scaling laws for LLMs. Vocabulary size has not dealt with in detail in prior work and this paper fill that gap. The study is rather limited in scale, but is perhaps understandable considering the very high computational demands that pretraining creates. It would be great if the study could be scaled up a little more, so that there is more confidence in the generalization.